# Consistent Plug-in Classifiers for Complex Objectives and Constraints

**Shiv Kumar Tavker**
Indian Institute of Technology Madras, India
shivtavker@smail.iitm.ac.in

**Harish G. Ramaswamy**
Indian Institute of Technology Madras, India
hariguru@cse.iitm.ac.in

**Harikrishna Narasimhan**
Google Research, USA
hnarasimhan@google.com

## Abstract

We present a consistent algorithm for constrained classification problems where the objective (e.g. F-measure, G-mean) and the constraints (e.g. demographic parity fairness, coverage) are defined by general functions of the confusion matrix. Our approach reduces the problem into a sequence of plug-in classifier learning tasks. The reduction is achieved by posing the learning problem as an optimization over the intersection of two sets: the set of confusion matrices that are achievable and those that are feasible. This decoupling of the constraint space then allows us to solve the problem by applying Frank-Wolfe style optimization over the individual sets. For objective and constraints that are convex functions of the confusion matrix, our algorithm requires $O(1/\epsilon^2)$ calls to the plug-in subroutine, which improves on the $O(1/\epsilon^3)$ calls needed by the reduction-based algorithm of Narasimhan (2018) [29]. We show empirically that our algorithm is competitive with prior methods, while being more robust to choices of hyper-parameters.

## 1 Introduction

In an increasing number of machine learning tasks, one is required to train a classifier with constraints on multiple metrics such as fairness, coverage, recall, etc [16, 17, 2, 9, 10]. Often, the objective and constraints in these problems are not simple metrics such as classification error, and may have a more complex non-decomposable structure, i.e. may not be expressible a simple average of errors on individual data points. Examples of such metrics include the F-measure and G-mean used in class-imbalanced problems [27, 24], the predictive parity criteria used in ML fairness [7], KL-divergence based metrics used in distribution matching tasks [12, 14], and many more.

A common feature of the above metrics is that they can all be defined as a function of a classifier's confusion matrix. We are therefore interested in constrained learning problems where the objectives and constraints are general functions of the confusion matrix. Our goal is to design a *statistically consistent* algorithm for solving these problems, i.e. an algorithm that converges in the limit of infinite training data to an optimal feasible classifier for these problems.

In previous work, Narasimhan (2018) [29] provide consistent algorithms for constrained learning problems by reducing them into a sequence of easy-to-solve sub-problems. Each of these sub-problems is a linear metric minimization task and involves learning a plug-in classifier, a classifier constructed by fine-tuning a threshold (or a weight matrix for multiclass problems) on a pre-trained class probability model. For convex functions of the confusion matrix, their method requires $O(1/\epsilon^3)$ calls to the plug-in learning routine to converge to a classifier that is $\epsilon$-optimal and $\epsilon$-feasible. In this

paper, we build on their work and provide an algorithm which requires only $O(1/\epsilon^2)$ calls to the plug-in routine to reach a classifier of the same quality.

Like the prior method, the key to our approach is to translate the constrained learning problem into an optimization problem over a finite dimensional space. While Narasimhan (2008) formulate this optimization problem over the space of confusion matrices that are achievable by a classifier, we formulate the problem over the intersection of *two* convex sets: the set of confusion matrices that are achievable, and the set of confusion matrices that are feasible, i.e. satisfy the constraints. The decoupling of the search space into two sets then allows us to adapt the Frank-Wolfe based algorithm of Gidel et al. (2018) [15] to solve the optimization. Our approach makes use of two oracle subroutines, both of which can be implemented efficiently. The first oracle minimizes a linear function over the space of achievable confusion matrices, which amounts to learning a plug-in classifier. The second performs a linear minimization over the space of feasible matrices, which is often a straight-forward convex program.

The proposed algorithm enjoys several practical benefits. Firstly, the algorithm is computationally efficient to implement: given a pre-trained class probability model (e.g. logistic regression), the algorithm performs a sequence of efficient threshold optimizations on the predicted class probability outputs. Secondly, it can be applied readily to multi-class problems and fairness problems with multiple groups. Thirdly, the number of optimization parameters needed by our algorithm scales linearly with the number of classes and groups, and does not directly depend of the number of constraints. This is in contrast to the method of Narasimhan (2018), which maintains an explicit parameter for each constraint. Our approach instead solves a linear minimization problem over the feasible matrices, which has the advantage of leveraging specialized convex solvers that exploit redundancies in the constraints.

**Contributions.** The following are the main contributions in this paper. (i) We provide an algorithm for complex constrained classification problems , which solves a sequence of plug-in learning tasks (see Section 3). (ii) We show that our algorithm is statistically consistent and enjoys improved convergence guarantees (see Section 4). (iii) We present experiments on benchmark fairness datasets and show that the proposed algorithm performs at least as well as existing methods, while being more robust to choices of hyper-parameters (see Section 5).

**Related Work.** Prior methods for optimizing complex evaluation metrics fall mainly under two broad categories: plug-in style methods that enjoy consistency guarantees [35, 25, 34, 33, 44, 3, 29], and approaches that optimize convex relaxations to the metrics and are not necessarily consistent [20, 22, 32, 21, 16, 37, 30, 19]. There has also been much work on training classifiers with objectives and constraints that are *linear* constraints on the confusion matrix, with the main focus being on fairness constraints [16, 46, 2, 23, 11, 9, 10, 31]. There's however been relatively lesser work on handling objectives and constraints that are non-linear in the confusion matrix [29, 30, 5]. The more recent of these approaches by Narasimhan et al. (2019) [30] formulates the constrained learning problem as a Lagrangian game played by three players, and seeks to find an equilibrium of the game. However, their main proposal makes use of "surrogate relaxations" for the entries of the confusion matrix and does not come with consistency guarantees. We compare against this algorithm in our experiments. Narasimhan et al. (2019) do however also provide a more idealized algorithm that enjoys the same convergence rate as our approach to the optimal feasible solution, but do not provide a consistency analysis for this method. In Section 4 and Appendix B, we discuss in detail about the technical differences between this idealized algorithm of theirs and our approach.

## 2 Preliminaries and Background

We are interested in general multiclass learning problems with an instance space $\mathcal{X}$ and label space $\mathcal{Y} = [n] = \{1, 2, \ldots, n\}$. For binary classification problems, we will denote the label space using $\mathcal{Y} = \{0, 1\}$. We use $\Delta_n$ to denote the probability simplex in $\mathbb{R}^n_+$. We assume examples are drawn i.i.d. from some distribution $D$ on $\mathcal{X} \times [n]$, with marginal $\mu$ on $\mathcal{X}$, conditional class probabilities $\eta_i(x) = \mathbf{P}(Y = i | X = x)$, and class priors $\pi_i = \mathbf{P}(Y = i)$. Given a finite training sample $S = ((x_1, y_1), ..., (x_N, y_N)) \in (\mathcal{X} \times [n])^N$ drawn i.i.d. from $D$, the task is to learn a multiclass classifier $h : \mathcal{X} \to [n]$, or more generally, a *randomized* multiclass classifier $h : \mathcal{X} \to \Delta_n$, which given an instance $x$ predicts a class label in $[n]$ according to the probability distribution specified by $h(x)$. Let $\mathcal{H}$ denote the the space of all randomized classifiers.

We will also be interested in fair classification problems where each instance belongs to one of $m$ protected groups, and will denote the protected group associated with instance $X$ by $A(X) \in [m]$. We denote $\nu_a = \mathbf{P}(A(X) = a)$ and $\pi_{a,i} = \mathbf{P}(A(X) = a, Y = i)$.

**Learning problem.** We measure the performance of a classifier w.r.t. distribution $D$ using a performance measure $\bar{\psi} : \mathcal{H} \to R_+$ that associates a non-negative value $\bar{\psi}(h; D) \in \mathbb{R}_+$ to each classifier $h \in \mathcal{H}$, with *lower* values indicating better performance. We also require the classifier to satisfy $K$ constraints, given by $\bar{\phi}_k(h; D) \le 0, k \in [K]$, where $\bar{\phi}_k : \mathcal{H} \to \mathbb{R}$ associates a real value to a classifier. Our goal is to then solve the following optimization problem over classifiers:

$$\min_{h \in \mathcal{H}} \bar{\psi}(h) \text{ s.t. } \bar{\phi}_k(h) \le 0, \forall k \in [K]. \tag{OP1}$$

**Confusion matrices.** We define the confusion matrix of a classifier $h$ as a $n \times n$ matrix $C[h] \in [0,1]^{n \times n}$ where the $ij$-the entry is the probability that the true class for an instance is $i$ and the predicted class is $j$:

$$C_{ij}[h] = \mathbf{P}_{Y,\widehat{Y} \sim h(X)}(Y = i, \widehat{Y} = j),$$

where $\widehat{Y} \sim h(X)$ denotes a random draw of label from $h(X)$. For fairness settings, we will also be interested in the group-specific confusion matrices:

$$C_{ij}^a[h] = \mathbf{P}_{X,Y,\widehat{Y} \sim h(X)}(Y = i, \widehat{Y} = j, A(X) = a)$$

**Complex objectives and constraints.** We will consider performance metrics $\bar{\psi}$ and constraint functions $\bar{\phi}_k$'s that are general functions of the confusion matrix of classifier $h$. This includes several common examples, including those that are *non-decomposable* and cannot be expressed as a simple expectation of errors on individual examples.

- *Class-imbalanced metrics* such as the G-mean, H-mean and Q-mean that emphasize equal performance acrossa all classes [27, 24, 39, 42, 26, 28] and metrics used in signal detection [41]:

$$\text{G-mean} = 1 - \left(\prod_{i=1}^n \frac{C_{ii}}{\pi_i}\right)^{1/n}; \quad \text{H-mean} = 1 - n \left(\sum_{i=1}^n \frac{\pi_i}{C_{ii}}\right)^{-1}$$

$$\text{Q-mean} = \sqrt{\frac{1}{n} \sum_{i=1}^n \left(1 - \frac{C_{ii}}{\pi_i}\right)^2}; \quad \text{Min-max} = \max_{i \in [n]} \left(1 - \frac{C_{ii}}{\pi_i}\right)$$

- *Fairness constraints* used to control the discrepancy in the performance of a classifier across different protected groups [17]:

$$\text{Demographic Parity: } \max_{a \in [m]} \left| \frac{1}{\nu_a}(C_{01}^a + C_{11}^a) - \frac{1}{m} \sum_{b=1}^m \frac{1}{\nu_b}\left(C_{01}^b + C_{11}^b\right) \right| \le \epsilon$$

$$\text{Equal Opportunity: } \max_{a \in [m]} \left| \frac{1}{\pi_{a,1}}C_{11}^a - \frac{1}{m} \sum_{b=1}^m \frac{1}{\pi_{b,1}}C_{11}^b \right| \le \epsilon,$$

where $\epsilon$ is an acceptable slack.
- *Coverage constraints* that require the proportion of predictions in a particular class to match a target value [16, 10, 8], and the related KL-divergence metric used in the quantification literature [12, 14, 21]:

$$\text{Binary Coverage: } C_{01} + C_{11} \le \epsilon$$

$$\text{KL-divergence: } \sum_{i=1}^n \pi_i \log \left(\frac{\pi_i}{\sum_{j=1}^n C_{ji}}\right) \le \epsilon.$$

**Confusion vectors.** For ease of presentation, we will work with a generalized version of a confusion matrix, which we refer to as a confusion vector. For a classifier $h$, we overload notation and define a confusion vector $C[h] \in \mathbb{R}^d$ as:

$$C_i[h] = \mathbf{E}_{X,Y}[\mathbf{E}_{\widehat{Y} \sim h(X)}[\sigma_i(X, Y, \widehat{Y})]],$$

for some *sufficient statistics* $\sigma_i : \mathcal{X} \times [n] \times [n] \to [0,1]$ computed on the instance $X$, true labels $Y$ and predicted labels $\widehat{Y}$. For example, when $\sigma_i(X, Y, \widehat{Y}) = \mathbf{1}(Y = i, \widehat{Y} = i)$, we get the diagonal elements of the standard confusion matrix with $d = n$. When we set $\sigma_i(X, Y, \widehat{Y}) = \mathbf{1}(Y = j, \widehat{Y} = k)$, we get back the $jk$-th entry of the standard confusion matrix, with the entire matrix can be represented by a $n^2$-dimensional confusion vector. When we set $\sigma_i(X, Y, \widehat{Y}) = \mathbf{1}(A(X) = a, Y = j, \widehat{Y} = k)$, we get back the $jk$-th entry of the group-specific confusion matrix for group $a$. The set of $m$ group-specific matrices can then be represented by a $mn^2$-dimensional confusion vector.

Figure 1: An illustration of Algorithm 1 for a toy 2-class problem, with equal prior probabilities and with class conditionals $X|Y=0$ and $X|Y=1$ distributed as a standard normal with means $+1$ and $-1$ respectively. The goal is to minimize H-mean subject to a coverage constraint that forces the fraction of class 1 predictions to be not more than $0.3$. The objective and constraint functions are given by: $\psi(C) = 1 - 2\left(\frac{0.5}{C_{00}} + \frac{0.5}{C_{11}}\right)^{-1}$ and $\phi(C) = C_{11} + C_{01} - 0.3$.

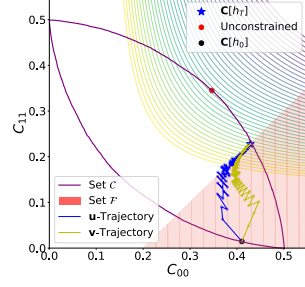

## 3 Reduction-based Algorithm

We now describe our approach for solving the learning problem in (OP1) by reducing the problem into a sequence of plug-in classifier learning tasks. We will work with objectives and constraints defined in terms of a confusion vector $C[h]$ of dimension $d$, for some suitable choice of sufficient statistics $\sigma_i$'s. Specifically, we consider an objective $\bar{\psi}(h) = \psi(C[h])$ defined by a *convex* function $\psi : [0,1]^d \to \mathbb{R}$ of the confusion vector for $h$, and constraint functions $\bar{\phi}_k(h) = \phi_k(C[h])$ defined by convex functions $\phi_k : [0,1]^d \to \mathbb{R}$ of the confusion vector for $h$.

### 3.1 Optimization Over Intersection of Convex Sets

Our key idea is to reformulate (OP1) as an optimization problem over the intersection of two convex sets. To this end, we define the set of all confusion vectors that can be achieved by some classifier $h$:

**Achievable Confusion Vectors**: $\mathcal{C} = \{\mathbf{u} \in \mathbb{R}^d : \mathbf{u} = C[h], h : \mathcal{X} \to \Delta_n\}$,

and the set of confusion vectors that satisfy the $K$ constraints:

**Feasible Confusion Vectors**: $\mathcal{F} = \{\mathbf{u} \in \mathbb{R}^d : \phi_k(\mathbf{u}) \leq 0, \forall k \in [K]\}$.

**Proposition 1.** $\mathcal{C}$ *and* $\mathcal{F}$ *are convex sets.*

The convexity of $\mathcal{C}$ follows from the use of randomised classifiers and the fact that $C[h]$ is defined as an expectation over random draw from $h$. The convexity of $\mathcal{F}$ follows from the convexity of the constraint functions $\phi_k$. Also notice that while the set of achievable confusion vectors $\mathcal{C}$ depends on the data distribution $D$, the set of feasible confusion vectors does not. This means that optimizing over $\mathcal{F}$ does not require access to $D$ or a sample drawn form $D$.

Equipped with these two sets, we can reformulate the learning problem in (OP1) over the space of classifiers, as an equivalent $d$-dimensional optimization problem over the intersection of $\mathcal{C}$ and $\mathcal{F}$:

$$\min_{\mathbf{u} \in \mathcal{C} \cap \mathcal{F}} \psi(\mathbf{u}). \tag{OP2}$$

We will denote the solutions to the problems (OP1) and (OP2) by $h^*$ and $\mathbf{u}^*$ respectively. Note that $C[h^*] = \mathbf{u}^*$. In Figure 1, we provide a simple illustration of an objective function and constraints on a toy problem, and show the corresponding sets $\mathcal{C}$ and $\mathcal{F}$.

### 3.2 Linear Minimization Oracles

The formulation (OP2) converts a classifier learning problem into a finite dimensional optimization problem, but it still has one major issue: *we do not have direct access to the set $\mathcal{C}$*. However, as we shall see shortly, performing a linear minimization over this set amounts to a cost-sensitive learning problem, which can be solved using a plug-in method. Similarly, performing a linear minimization over $\mathcal{F}$ amounts to solving a convex program.

So, we assume access to the following linear minimization oracles (LMOs):

$$\text{LMO}_{\mathcal{C}}: \text{Given } \mathbf{a} \in \mathbb{R}^d, \text{ returns } \operatorname*{argmin}_{\mathbf{u} \in \mathcal{C}} \langle \mathbf{a}, \mathbf{u} \rangle,$$
$$\text{LMO}_{\mathcal{F}}: \text{Given } \mathbf{b} \in \mathbb{R}^d, \text{ returns } \operatorname*{argmin}_{\mathbf{v} \in \mathcal{F}} \langle \mathbf{b}, \mathbf{v} \rangle.$$

---

**Algorithm 1** The Split Bayes-Frank-Wolfe (SBFW) Algorithm

---

1: **Input:** $\psi : [0,1]^d \to \mathbb{R}_+$, Linear minimization oracle over $\mathcal{F}$
        Training sample $S = \{(x_1, y_1), \ldots, (x_N, y_N)\}$
2: **Parameters:** $\lambda > 0$, Step sizes $\eta_t = C/t$, and $\gamma_t = \frac{4\eta_t}{\lambda}$ for $t \in [T]$, where $C$ is some constant.
3: **Initialize:** Initialize classifier $h_0 : \mathcal{X} \to \Delta_n$ and vectors $\mathbf{u}_0 = \mathbf{v}_0 = C[h_0]$, $\mathbf{w}_0 = 0$.
4: **For** $t = 1$ **to** $T$ **do:**
5:     $\widehat{g}_t, \widetilde{\mathbf{u}}_t = \text{plug-in}(\mathbf{a}_{t-1}; S)$, where $\mathbf{a}_{t-1} = \nabla_{\mathbf{u}} \mathcal{L}(\mathbf{u}_{t-1}, \mathbf{v}_{t-1}, \mathbf{w}_{t-1})$     (LMO over $\mathcal{C}$)
6:     $\widetilde{\mathbf{v}}_t = \text{argmin}_{\mathbf{v} \in \mathcal{F}} \langle \mathbf{b}_{t-1}, \mathbf{v} \rangle$, where $\mathbf{b}_{t-1} = \nabla_{\mathbf{v}} \mathcal{L}(\mathbf{u}_{t-1}, \mathbf{v}_{t-1}, \mathbf{w}_{t-1})$     (LMO over $\mathcal{F}$)
7:     $(\mathbf{u}_t, \mathbf{v}_t, h_t) = (1 - \gamma_t)(\mathbf{u}_{t-1}, \mathbf{v}_{t-1}, h_{t-1}) + \gamma_t(\widetilde{\mathbf{u}}_t, \widetilde{\mathbf{v}}_t, \widehat{g}_t)$
8:     $\mathbf{w}_t = \mathbf{w}_{t-1} + \eta_{t-1}(\mathbf{u}_t - \mathbf{v}_t)$
9: **end For**
10: **Return:** Return $\widehat{h} = h_{\text{Best}}$, where Best $= \text{argmin}_{t > T/2} \|\mathbf{u}_t - \mathbf{v}_t\|^2$

---

Of the two oracles, $\text{LMO}_{\mathcal{F}}$ does not need access to the data and can performed with standard convex solvers. So, we will be primarily interested in the number of calls needed to be made to $\text{LMO}_{\mathcal{C}}$. Also note that in practice, one may not be able to solve the minimization over $\mathcal{C}$ exactly. In our theoretical analysis in Section 4, we take this into account and show that our approach is robust to approximation errors in the linear minimization.

### 3.3 Frank-Wolfe Based Algorithm

The challenge now is to optimize over the intersection of the two sets $\mathcal{C} \cap \mathcal{F}$. For this, we adopt the Frank-Wolfe based approach of Gidel et al. (2018) [15] that enables optimization of a convex objective over the intersection of two convex sets with access to only linear minimization oracles for the individual sets. To this end, we introduce auxiliary variables $\mathbf{v}$ in (OP2) and decouple the two constraint sets. This gives us the following equivalent optimization problem:

$$\min_{\mathbf{u} \in \mathcal{C}, \mathbf{v} \in \mathcal{F}} \psi(\mathbf{u}) + \psi(\mathbf{v}) \text{ s.t. } \mathbf{u} - \mathbf{v} = 0. \tag{OP3}$$

We then define the augmented Lagrangian $\mathcal{L} : [0,1]^d \times [0,1]^d \times \mathbb{R}^d \to \mathbb{R}$ of the above problem as:

$$\mathcal{L}(\mathbf{u}, \mathbf{v}, \mathbf{w}) = \psi(\mathbf{u}) + \psi(\mathbf{v}) + \mathbf{w}^\top(\mathbf{u} - \mathbf{v}) + \frac{\lambda}{2}\|\mathbf{u} - \mathbf{v}\|^2, \tag{1}$$

where $\mathbf{w} \in \mathbb{R}^d$ denotes the Lagrange multipliers for the equality constraints and $\lambda > 0$ is a constant.

Gidel at al. (2018) [15] propose a simple gradient ascent step for $\mathbf{w}$, a linear minimization step for $\mathbf{u}$ over $\mathcal{C}$ and a linear minimization step for $\mathbf{v}$ over $\mathcal{F}$. Specifically, at each iteration, we perform a Frank-Wolfe style update for $\mathbf{u}$ and $\mathbf{v}$ [18]. We linearize the Lagrangian with respect to $\mathbf{u}$ and minimize the linearized objective over $\mathcal{C}$ using $\text{LMO}_{\mathcal{C}}$:

$$\mathbf{a}_{t-1} = \nabla_{\mathbf{u}} \mathcal{L}(\mathbf{u}_{t-1}, \mathbf{v}_{t-1}, \mathbf{w}_{t-1}); \quad \widetilde{\mathbf{u}}_t \in \text{argmin}_{\mathbf{u} \in \mathcal{C}} \langle \mathbf{a}_{t-1}, \mathbf{u} \rangle. \tag{2}$$

We also linearize $\mathcal{L}$ with respect to $\mathbf{v}$ and minimize the linearized objective over $\mathcal{F}$ using $\text{LMO}_{\mathcal{F}}$:

$$\mathbf{b}_{t-1} = \nabla_{\mathbf{v}} \mathcal{L}(\mathbf{u}_{t-1}, \mathbf{v}_{t-1}, \mathbf{w}_{t-1}); \quad \widetilde{\mathbf{v}}_t \in \text{argmin}_{\mathbf{v} \in \mathcal{F}} \langle \mathbf{b}_{t-1}, \mathbf{v} \rangle. \tag{3}$$

This is followed by a set of simple updates on the optimization variables:

$$\mathbf{u}_t = (1 - \gamma_t)\mathbf{u}_{t-1} + \gamma_t \widetilde{\mathbf{u}}_t; \qquad \mathbf{v}_t = (1 - \gamma_t)\mathbf{v}_{t-1} + \gamma_t \widetilde{\mathbf{v}}_t; \tag{4}$$

$$\mathbf{w}_t = \mathbf{w}_{t-1} + \eta_{t-1}(\mathbf{u}_t - \mathbf{v}_t), \tag{5}$$

where the coefficients $\gamma_t$ and $\eta_t$ are step-size parameters. The procedure outlined in Algorithm 1 maintains both a confusion vector and the corresponding classifier at each iteration, and returns a classifier $\widehat{h}$ that combines multiple classifiers via randomization.

### 3.4 Plug-in Classifier for LMO over $\mathcal{C}$

All that remains is to perform the linear minimization over $\mathcal{C}$ in Equation 2. We show below that this can be solved using a plug-in method.

---

**Algorithm 2** Plug-in Method for LMO$_\mathcal{C}$

---

1: **Input:** Weight vector $\mathbf{a} \in \mathbb{R}^d$, Training sample $S = \{(x_1, y_1), \ldots, (x_N, y_N)\}$
2: **Given:** A conditional probability model $\widehat{\eta} : \mathcal{X} \to \Delta_n$ pre-trained with samples $\{(x_i, y_i)\}_{i=1}^{N/2}$, Sufficient statistic functions $\sigma_i : \mathcal{X} \times [n] \times [n] \to [0, 1]$
3: Define $\mathbf{L} : \mathcal{X} \to \mathbb{R}^{n \times n}$ by $L_{j,k}(x) = \sum_{i=1}^d a_i \sigma_i(x, j, k)$
4: Construct $\widehat{g} : \mathcal{X} \to [n]$ as $\widehat{g}(x) = \operatorname{argmin}_{\widehat{y} \in [n]} \sum_{j=1}^n \widehat{\eta}_j(x) \, L_{j,\widehat{y}}(x)$,
5: Estimate confusion vector $\widetilde{u}_i = \frac{2}{N} \sum_{j=N/2}^N \sigma_i(x_j, y_j, \widehat{g}(x_j))$ from samples $\{(x_i, y_i)\}_{i=N/2}^N$
6: **Return:** Confusion vector $\widetilde{\mathbf{u}}$ and corresponding classifier $\widehat{g}$

---

**Proposition 2** (**LMO$_\mathcal{C}$ through Bayes-optimal Classifier**). *Suppose we wish to minimize* $\langle \mathbf{a}, \mathbf{u} \rangle$ *over* $\mathbf{u} \in \mathcal{C}$. *Define the example-dependent loss matrix* $\mathbf{L} : \mathcal{X} \to \mathbb{R}^{n \times n}$ *as* $L_{j,k}(x) = \sum_{i=1}^d a_i \sigma_i(x, j, k)$. *Then the solution to the linear minimization problem is directly given by the Bayes-optimal classifier for this loss matrix. Specifically, construct a classifier* $g^* : \mathcal{X} \to [n]$ *with*

$$g^*(x) = \operatorname*{argmin}_{\widehat{y} \in [n]} \sum_{j=1}^n \eta_j(x) \, L_{j,\widehat{y}}(x),$$

*where* $\eta_j(x) = \mathbf{P}(Y = 1 | x)$ *is the class-conditional probability. Then* $C[g^*] \in \operatorname{argmin}_{\mathbf{u} \in \mathcal{C}} \langle \mathbf{a}, \mathbf{u} \rangle$.

The classifier $g^*$ defined above is a deterministic classifier that thresholds the conditional probability $\eta$ based on the example-dependent loss matrix $\mathbf{L}(x)$. For the special case where the confusion vectors represent the set of confusion matrices for the $m$ groups, the weight vector $\mathbf{a} \in \mathbb{R}^{mn^2}$ effectively describes $m$ loss matrices $\mathbf{L}^1, \ldots, \mathbf{L}^m \in \mathbb{R}^{n \times n}$, one for each group. For a given instance $x$, the classifier $g^*$ picks the loss matrix $\mathbf{L}^{A(x)}$ associated with the protected group attribute $A(x)$, and then uses the conditional probability vector $\eta(x)$ to make the optimal prediction for that loss matrix: $g^*(x) = \operatorname{argmin}_{\widehat{y} \in [n]} \sum_{j=1}^n \eta_j(x) L_{j,\widehat{y}}^{A(x)}(x)$.

The above characterization directly motivates the use of a plug-in method to solve the LMO over $\mathcal{C}$. Specifically, we can use an estimator $\widehat{\eta} : \mathcal{X} \to \Delta_n$ of the conditional probabilities $\eta$ to construct an approximate version of $g^*$. The confusion vector $C[g^*]$ can then be estimated from samples. This procedure is outlined in Algorithm 2 and returns both a confusion vector that approximately solves the linear minimization over $\mathcal{C}$ and the corresponding classifier $\widehat{g}$. Notice that the conditional probability estimator $\widehat{\eta}$ (e.g. logistic regression) needs to be trained only once, and can be re-used for every call to the plug-in routine.

Figure 1 shows the iterates of the proposed algorithm over a simple toy dataset. The trajectory of $\mathbf{u}_t$ is given in blue and the trajectory of $\mathbf{v}_t$ is given in yellow. It can be seen that both these trajectories approach the optimal solution $C[h^*]$.

# 4  Consistency Results

In this section we give the main theoretical result of the paper. We show that with $O(1/\epsilon^2)$ calls to the plug-in LMO routine, Algorithm 1 outputs a classifier $\widehat{h}$ that is $O(\epsilon + \sqrt{\rho})$-close to the optimal-$\psi$ value and satisfies the constraint $\phi_k$'s with a slack of $O(\epsilon + \sqrt{\rho})$, where $\rho$ is a term which depends on the approximation level of the plug-in LMO. This result then directly implies that Algorithm 1 is statistically consistent, i.e. converges to the optimal-feasible classifier in the limit of infinite samples.

We will make a few regularity assumptions. We assume that the objective function $\psi$ and constraint functions $\phi_k$ are $L$-Lipschitz and objective function $\psi$ is $\beta$-smooth. We will also assume that (OP2) is strictly feasible.

**Assumption 1.** $\exists \mathbf{u} \in \mathcal{C} \cap \mathcal{F}, r > 0$ *such that* $B(\mathbf{u}, r) \cap$ *affine-space*$(\mathcal{C}) \subseteq \mathcal{C} \cap \mathcal{F}$.

We stress that these assumptions are not very restrictive and can be verified to be satisfied by all of the objectives and constraints described in Section 2, as long as the prior probabilities $\pi_{a,i}$ are non-zero for all classes $i \in [n]$ and protected groups $a \in [m]$.

**Theorem 3.** *Let $h^*$ denote the optimal feasible solution for (OP1), i.e. $\phi_k(C[h^*]) \leq 0, \forall k$ and $\psi(C[h^*]) \leq \psi(C[h])$ for all $h$ that is feasible. Under the regularity assumptions, for large enough $\lambda$ and an appropriate step-size parameter $C$, there exists an $\bar{\epsilon} > 0$ such that, for all $\epsilon \leq \bar{\epsilon}$, and $T \geq \dfrac{c}{\epsilon^2}$, with probability $1 - \delta$ over draw of the training samples $S$ i.i.d. from $D$, the classifier $\widehat{h}$ returned by Algorithm 1 is near-optimal and near-feasible:*

$$\textbf{\textit{Optimality}} : \quad \psi(C[\widehat{h}]) \leq \psi(C[h^*]) + c\sqrt{\rho} + \epsilon,$$

$$\textbf{\textit{Feasibility}} : \quad \phi_k(C[\widehat{h}]) \leq c\sqrt{\rho} + L\epsilon, \, \forall k \in [K],$$

*where $\rho = \sqrt{d}\mathbf{E}\|\eta(X) - \widehat{\eta}(X)\|_1 + d\sqrt{\dfrac{d\log(d) + \log(Nn^2) + \log(1/\delta)}{N}}$ captures the approximation level of the LMO given by Algorithm 2, and $c > 0$ is a constant not dependent on the number of iterations $T$ and the training samples.*

The key to proving this convergence result is (i) establishing that the plug-in classifier solves the linear minimization problem over $\mathcal{C}$ approximately, (ii) applying the convergence results of Gidel et al. (2018) [15] (extended to handle an approximate LMO) to get a bound on the duality gap for (OP2), and (iii) translating this to a bound on the optimality and feasibility for (OP2).

**Remark (Consistency).** The term $\rho$ in Theorem 3 has two sources of error: the error $\mathbf{E}\|\eta(X) - \widehat{\eta}(X)\|_1$ in the class probability model $\widehat{\eta}$ used to construct the plug-in classifier and the sample error $\widetilde{\mathcal{O}}\left(d\sqrt{\dfrac{d}{N}}\right)$. If the conditional-class estimator is such that $\mathbf{E}\|\eta(X) - \widehat{\eta}(X)\|_1 \to 0$ as the sample size $N \to \infty$, which is the case when e.g. $\widehat{\eta}$ is learned by minimizing a strictly proper composite loss over a suitably flexible function class [40], then Algorithm 1 is guaranteed to be statistically consistent. Specifically, setting $\epsilon = \sqrt{1/N}$ and running Algorithm 1 for the prescribed $O(1/\epsilon^2)$ iterations, we have that as $N \to \infty$, $\psi(C[\widehat{h}]) \xrightarrow{P} \psi(C[h^*])$ and $\phi_k(C[\widehat{h}]) \xrightarrow{P} 0, \forall k$.

**Remark (Improvements over COCO [29]).** The previous reduction-based algorithm of Narasimhan (2018) [29] for (OP1), referred to as COCO by the author, similarly poses the problem as an optimization over $\mathcal{C}$ but retains explicit constraints $\phi_k(C) \leq 0, \forall k$. The idea is to then formulate the Lagrangian for the constrained problem with one Lagrange multiplier for each constraint, and maximize the Lagrangian over the multipliers using gradient ascent. Each gradient step, however, involves a full run of the classical Frank-Wolfe method [18] over $\mathcal{C}$ using an LMO, resulting in an algorithm with multiple levels of nesting. Our approach is better than COCO in the following aspects:

- *Better convergence rate.* In the large $N$ setting, COCO requires $O(1/\epsilon^3)$ calls to the plug-in routine to reach a solution that is $O(\epsilon)$-optimal and $O(\epsilon)$-feasible. In contrast, by posing (OP1) as an optimization over two convex sets, we avoid the nested structure, and need only $O(1/\epsilon^2)$ calls to the plug-in routine to reach a solution of the same quality.
- *Weaker dependence on the number of constraints.* While COCO maintains one optimization parameter per constraint, the number of parameters in our algorithm (i.e. $\mathbf{u}, \mathbf{v}$) is only twice the dimension $d$ of the confusion vector, and depends on the number of constraints $K$ only through the LMO over $\mathcal{F}$. This has the added advantage of being able to use specialized solvers for this step that better exploit the redundancies in the constraint set.

**Remark (Prior 3-player approach [30]).** As noted in the introduction, another closely related method for solving complex constrained classification problems is the 3-player approach of Narasimhan et al. (2019) [30]. Their idea is to introduce additional slack variables, formulate the Lagrangian for the problem with one parameter per constraint, and find an equilibrium of the resulting min-max game between the primal and dual variables. They first provide an idealized version of their algorithm which makes use of an oracle (similar to $\text{LMO}_\mathcal{C}$) to optimize a linear metric over the space of classifiers, and requires a similar number of calls to the oracle as our approach to reach a near-optimal near-feasible solution. However, they do not provide a full-fledged consistency analysis for this idealized algorithm. Instead they prescribe a "practical" alternative which replaces the oracle with stochastic gradient updates on a relaxed Lagrangian, where the entries of the confusion matrix are replaced with surrogate relaxations, and this variant does not come with consistency guarantees. We compare with this surrogate-based algorithm in our experiments. Again, an important difference between our approach and Narasimhan et al. (2019) is that we do not maintain an explicit parameter for each constraint and access the constraint set only through an LMO.

Table 1: Minimizing Q-mean s.t. Demographic Parity $\leq 0.05$. We report test Q-mean and constraint violations (in parentheses) measured as the positive part of Demographic Parity $- 0.05$. *Lower* values are better. **Bold** indicates that the method has the least objective and the least violation, among the last three columns.

| Dataset | Unconstrained | Error-Con | COCO | 3-Player | Proposed |
|---------|---------------|-----------|------|----------|----------|
| Adult | 0.18 (0.05) | 0.30 (0.00) | 0.31 (0.00) | **0.18 (0.00)** | **0.18 (0.00)** |
| COMPAS | 0.32 (0.10) | 0.36 (0.00) | 0.35 (0.03) | 0.33 (0.00) | **0.32 (0.00)** |
| Crimes | 0.16 (0.22) | 0.30 (0.01) | 0.30 (0.01) | 0.24 (0.05) | **0.22 (0.03)** |
| Default | 0.33 (0.01) | 0.54 (0.00) | 0.35 (0.00) | 0.36 (0.00) | **0.33 (0.00)** |
| Lawschool | 0.21 (0.25) | 0.47 (0.00) | 0.35 (0.16) | 0.24 (0.03) | 0.25 (0.02) |

Table 2: Minimizing G-mean s.t. Equal Opportunity $\leq 0.05$. We report G-mean and constraint violations measured as the positive part of Equal Opportunity $- 0.05$. *Lower* values are better.

| Dataset | Unconstrained | Error-Con | COCO | 3-Player | Proposed |
|---------|---------------|-----------|------|----------|----------|
| Adult | 0.18 (0.00) | 0.24 (0.01) | 0.17 (0.03) | 0.18 (0.01) | 0.18 (0.00) |
| COMPAS | 0.32 (0.09) | 0.35 (0.00) | **0.32 (0.00)** | 0.33 (0.00) | **0.32 (0.00)** |
| Crimes | 0.15 (0.17) | 0.19 (0.09) | 0.16 (0.06) | 0.16 (0.08) | **0.16 (0.03)** |
| Default | 0.33 (0.00) | 0.51 (0.00) | 0.39 (0.00) | 0.36 (0.00) | **0.34 (0.00)** |
| Lawschool | 0.21 (0.23) | 0.47 (0.00) | 0.23 (0.00) | 0.22 (0.04) | 0.26 (0.03) |

We provide more details about the prior COCO and 3-player methods in Appendix B.

## 5  Experiments

We show that the proposed algorithm performs comparable to or better than than prior methods for constrained classification on a number of benchmark datasets for fair classification.

**Datasets.** We ran experiments on five datasets: (1) *COMPAS*, where the goal is to predict recidivism with *gender* as the protected attribute [4]; (2) *Communities & Crime*, where the goal is to predict if a community in the US has a crime rate above the 70th percentile [13], and we consider communities having a black population above the 50th percentile as protected [23]; (3) *Law School*, where the task is to predict whether a law school student will pass the bar exam, with *race* (black or other) as the protected attribute [43]; (4) *Adult*, where the task is to predict if a person's income exceeds 50K/year, with *gender* as the protected attribute [13]; (5) *Default*, where the task is to predict if a credit card user defaulted on a payment, with gender as the protected attribute [13]. The details are summarized in Table 4 in Appendix C. We used 2/3-rd of the data for training and 1/3-rd for testing. All experiments use a linear model.[1]

**Comparisons.** We compare our method against (i) the approach of optimizing the given objective without constraints [33] (Unconstrained), (ii) the approach of optimizing classification error subject to the given constraints, e.g. [1] (Error-Con), (iii) the prior COCO method [29] for solving the constrained learning problem at hand, and (iv) the 3-player approach [30] which solves the constrained learning problem with surrogates. We describe how we choose hyper-parameters in Appendix C

**Objectives and Constraints.** We consider the following constrained learning tasks:

1. Minimizing Q-mean s.t. Demographic Parity Violation $\leq 0.05$
2. Minimizing G-mean s.t. Equal Opportunity Violation $\leq 0.05$
3. Minimizing H-mean s.t. Coverage for Class 1 $\leq 0.25$

We report the objectives and constraint violations (the positive part of $\phi(h) - \epsilon$) for the different methods in Tables 1–3. On a majority of the datasets, the proposed method is able to closely satisfy the constraints while achieving comparable or better objectives. As expected, unconstrained optimization of the objective performs poorly on the constraints. Similar, optimizing for plain error rate subject to the specified constraints fares poorly on the desired objective, demonstrating the need to directly optimize for the metric one cares about. Among the SBFW (proposed), COCO and 3-Player methods, our approach is able to more often achieve the least objective and the least violation.

Table 3: Minimizing H-mean s.t. Class-1 Coverage $\leq 0.25$. We report etst H-mean and constraint violations measured as the positive part of Coverage $- 0.25$. *Lower* values are better. **Bold** indicates that the method has the least objective and the least violation, among last three columns.

| Dataset | Unconstrained | Error-Con | COCO | 3-Player | Proposed |
|---------|---------------|-----------|------|----------|----------|
| Adult | 0.18 (0.09) | 0.26 (0.00) | **0.21 (0.00)** | **0.21 (0.00)** | **0.21 (0.00)** |
| COMPAS | 0.32 (0.23) | 0.44 (0.00) | 0.45 (0.00) | 0.45 (0.00) | **0.44 (0.00)** |
| Crimes | 0.16 (0.11) | 0.21 (0.01) | 0.21 (0.01) | 0.23 (0.00) | 0.21 (0.01) |
| Default | 0.33 (0.16) | 0.62 (0.00) | **0.34 (0.00)** | 0.42 (0.00) | **0.34 (0.00)** |
| Lawschool | 0.21 (0.47) | 0.56 (0.01) | 0.58 (0.00) | 0.56 (0.00) | 0.55 (0.01) |

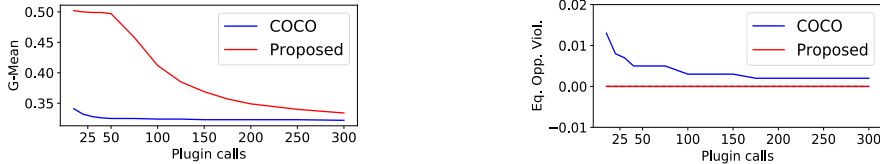

Figure 2: Training G-mean (left) and equal opportunity violation (right) on COMPAS for varying number of calls to the plug-in routine. The hyper-parameters were tuned separately for each method using the heuristic of Cotter et al. (2019) [10] to trade-off between the objective and the violations.

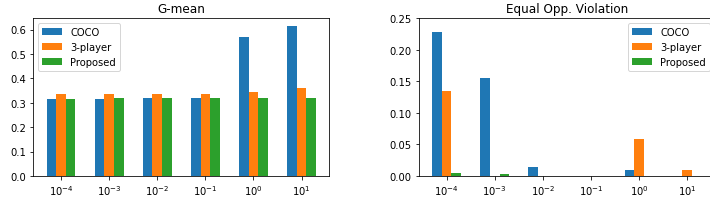

Figure 3: Robustness to hyper-parameters: Train G-mean and equal opportunity violation for six step sizes (*lower* is better) on the COMPAS dataset. For the proposed algorithm, all choices achieved similar objectives and near-zero violations.

**Convergence Analysis.** We next compare the number of plug-in calls needed by the proposed algorithm and the previous COCO method for the task of minimizing G-mean with an equal opportunity constraint. The 3-player method does not use a plug-in subroutine. Figure 2 shows the train G-mean and the train equal opportunity violation (the positive part of $\phi(h) - \epsilon$) for varying numbers of plug-in calls for the COMPAS dataset. In this case, our algorithm converges to a classifier with zero violation on the training set, with an objective similar to COCO, but with fewer calls ($\leq 100$). We also provide similar plots for other datasets in Figure 4 in Appendix C. On Crimes and Law School, COCO fails to converge to zero training violation even after 2000 calls. In contrast, on all five datasets, when provided the same number of plug-in calls, the proposed algorithm is able to achieve zero training violations (often within the first 100 calls). On Adult alone, COCO exhibits faster convergence.

**Robustness to Hyper-parameter Choices.** In our final experiment, we demonstrate the robustness of our approach to the choice of step-size $\eta_t$. We ran COCO, 3-player and the proposed SBFW methods for minimizing G-mean objective with an equal opportunity constraint on the COMPAS dataset, with 6 different choices of step-sizes ($10^{-4}, 10^{-3}, \ldots, 10$), and report the G-mean and equal opportunity violation in Figure 3 (and also as a scatter plot in Figure 5 in the Appendix). While all 6 choices achieved close-to-best objectives and near-zero violations for the proposed SBFW algorithm, only 2 (3 resp.) choices led to similar metrics for COCO (3-player resp.).

## 6   Conclusion

In numerous real-word prediction tasks, one is required to learn a classifier that optimizes a complex evaluation metric subject to a set of constraints. In this paper, we developed a consistent learning algorithm for handling objectives and constraints that are convex functions of the confusion matrix and provided improved convergence guarantees. In our experiments, we demonstrated the effectiveness of our approach, and also showed its robustness to hyper-parameter choices. In the future, it would be interesting to explore lower bounds on the number of calls to the LMO, replace the plug-in

LMO routine with more direct cost-sensitive learning methods (e.g. [38, 45]), and explore other optimization methods in place of the augmented Lagrangian Frank-Wolfe algorithm.

## Broader Impact

There's an increasing impetus in the machine learning community to design algorithms that are fair and free from bias and inequity. Most existing approaches for enforcing group-based fairness goals have been limited to simple objectives and constraints. In this paper, we allow a user to specify for more nuanced definitions of utilities and fairness goals than allowed by standard methods in the literature, and provide an algorithm to directly and efficiently optimize for these goals. We show theoretically that our algorithm is able to achieve a desired trade-off between overall utility and the specified fairness criteria.

As with prior work on group-based fairness (and more generally with constrained supervised learning), a drawback of our approach is that while we guarantee that the fairness criterion is likely to be satisfied on new examples, there is a small probability that it isn't, and these rare failures can have an adverse impact in practice. Moreover, our algorithm requires the use of stochastic classifiers, which may bring in additional ethical considerations. See Cotter et al. [8] for a discussion on the practical ramifications of deploying a stochastic classifier, and for ways to convert a stochastic classifier into a similar performing deterministic classifier.

All experiments in this paper were carried out with publicly available datasets.

## Acknowledgments and Disclosure of Funding

SKT and HGR thank the Robert Bosch Center for Data Science and Artificial Intelligence at IIT Madras for their support. The authors thank Shivani Agarwal for helpful discussions.

## Footnotes

[1]Code available at: `https://github.com/shivtavker/constrained-classification`.

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
