[Supplementary Material]

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

# A Proofs

Before we give the proofs, we define some terms necessary for the proofs and make the assumptions on the problem more explicit.

## A.1 Proof Setup

### A.1.1 Problem Assumptions

We had made several assumptions on the problem in the paper, which we recall here for reference.

1. The sufficient statistics functions $\sigma_1, \ldots, \sigma_d$ are bounded between $0$ and $1$.

2. The functions $\psi, \phi_k$ are convex.

3. The function $\psi : [0, 1]^d \to \mathbb{R}$ is also bounded between $0$ and $R$

4. The functions $\psi$ and $\phi_k$ are $L$-Lipschitz, i.e. $|\psi(\mathbf{u}) - \psi(\mathbf{u}')| \leq L||\mathbf{u} - \mathbf{u}'||_2$ and $|\phi_k(\mathbf{u}) - \phi_k(\mathbf{u}')| \leq L||\mathbf{u} - \mathbf{u}'||_2$

5. The function $\psi$ is $\beta$-smooth, i.e. $||\nabla\psi(\mathbf{u}) - \nabla\psi(\mathbf{u}')||_2 \leq \beta||\mathbf{u} - \mathbf{u}'||_2$

6. The sets $\mathcal{C} \subseteq [0, 1]^d$ and $\mathcal{F} \subseteq \mathbb{R}^d$ are full-dimensional, i.e. interiors are not empty.

7. The interiors of the sets $\mathcal{C}$ and $\mathcal{F}$ intersect. For some $r > 0$, there exists $\mathbf{c} \in \mathcal{C} \cap \mathcal{F}$ such that $B(\mathbf{c}, r) \subseteq \mathcal{C} \cap \mathcal{F}$.

The last two assumptions are made only for convenience and can be relaxed to Assumption 1. Any problem for which $\mathcal{C}$ and $\mathcal{F}$ are not full dimensional, can be converted to an equivalent problem where they are full dimensional by projecting the sufficient statistic functions $\sigma$ on to an appropriate affine space. Note that as the sufficient statistic functions take values $[0, 1]$, the set $\mathcal{C}$ is always a subset of $[0, 1]^d$, and we can also assume without loss of generality $\mathcal{F} \subseteq [0, 1]^d$.

In the proofs of the Theorems below, we will use $c$ to denote constants independent of the LMO error $\rho$ and number of iterations $T$. The value of $c$ can change even in consecutive expressions, to avoid cluttering the proof with unnecessary subscripts.

### A.1.2 Extra Definitions

**Definition 1** (Linear Minimization Oracle). *Let $\rho, \rho', \delta \in (0, 1)$. A linear minimization oracle, denoted by $\Omega$, takes a loss vector $\mathbf{a} \in \mathbb{R}^d$ and a sample $S$ as input, and outputs a classifier $\widehat{g}$ and an estimate of its confusion vector $\widetilde{\mathbf{u}} \in \mathbb{R}^d$. We say the $\Omega$ is a $(\rho, \rho', \delta)$-approximate LMO for sample size $N$ if for all $\mathbf{a} \in \mathbb{R}^d$, it outputs $(\widehat{g}, \widetilde{\mathbf{u}}) = \Omega(\mathbf{a}; S)$ such that:*

$$\langle \mathbf{a}, \mathbf{C}[\widehat{g}] \rangle \leq \min_{h:\mathcal{X}\to\Delta_n} \langle \mathbf{a}, \mathbf{C}[h] \rangle + \rho'\|\mathbf{a}\|$$

$$\|\mathbf{C}[\widehat{g}] - \widetilde{\mathbf{u}}\| \leq \rho.$$

*where the second inequality above is only required to hold with probability $1 - \delta$ over the sample $S$. The approximation constants $\rho$ and $\rho'$ may in turn depend on the sample size $N$, the dimension $d$ and the confidence level $\delta$.*

**Definition 2** (Fat Achievable Set). *The set $\mathcal{C}_\rho$ is defined as follows:*

$$\mathcal{C}_\rho = \mathcal{C} + B(\mathbf{0}, \rho) = \{\mathbf{u} + \mathbf{r} : \mathbf{u} \in \mathcal{C}, \mathbf{r} \in B(\mathbf{0}, \rho)\}$$

**Definition 3** (Augmented Lagrangian). *The Augmented Lagrangian $\mathcal{L} : \mathbb{R}^d \times \mathbb{R}^d \times \mathbb{R}^d \to \mathbb{R}$ is defined as*

$$\mathcal{L}(\mathbf{u}, \mathbf{v}, \mathbf{w}) = \psi(\mathbf{u}) + \psi(\mathbf{v}) + \frac{\lambda}{2}\|\mathbf{u} - \mathbf{v}\|^2 + \mathbf{w}^\top(\mathbf{u} - \mathbf{v})$$

Simple algebra shows that $\mathcal{L}(., ., \mathbf{w})$ is convex, Lipschitz continuous and smooth. We will require the following related inequalities for our Theorems.

**Proposition 4.** *For all $\mathbf{w} \in \mathbb{R}^d$, we have*

$$|\psi(\mathbf{u}) + \psi(\mathbf{v}) - \psi(\mathbf{u}') - \psi(\mathbf{v}')| \leq 2L\sqrt{\|\mathbf{u} - \mathbf{u}'\|^2 + \|\mathbf{v} - \mathbf{v}'\|^2}$$

$$\|\nabla_\mathbf{u}\mathcal{L}(\mathbf{u}, \mathbf{v}, \mathbf{w}) - \nabla_\mathbf{u}\mathcal{L}(\mathbf{u}', \mathbf{v}', \mathbf{w})\| \leq \beta_\lambda\|[\mathbf{u} - \mathbf{u}', \mathbf{v} - \mathbf{v}']\|$$

$$\|\nabla_\mathbf{v}\mathcal{L}(\mathbf{u}, \mathbf{v}, \mathbf{w}) - \nabla_\mathbf{v}\mathcal{L}(\mathbf{u}', \mathbf{v}', \mathbf{w})\| \leq \beta_\lambda\|[\mathbf{u} - \mathbf{u}', \mathbf{v} - \mathbf{v}']\|$$

*where we use $\nabla_\mathbf{u}$ and $\nabla_\mathbf{v}$ to denote the gradient w.r.t. the first and second arguments of $\mathcal{L}$, and $\beta_\lambda = 2\beta + 2\lambda$.*

**Definition 4** (Dual Function). *The dual function $\xi : \mathbb{R}^d \to \mathbb{R}$ is defined as*

$$\xi(\mathbf{w}) = \min_{\mathbf{u} \in \mathcal{C}_\rho, \mathbf{v} \in \mathcal{F}} \mathcal{L}(\mathbf{u}, \mathbf{v}, \mathbf{w})$$

We also use $\widehat{\mathbf{u}}(\mathbf{w}), \widehat{\mathbf{v}}(\mathbf{w})$ to denote any arbitrary minimizer of $\mathcal{L}(.,.,\mathbf{w})$ over $\mathcal{C}_\rho \times \mathcal{F}$. Thus $\xi(\mathbf{w}) = \mathcal{L}(\widehat{\mathbf{u}}(\mathbf{w}), \widehat{\mathbf{v}}(\mathbf{w}), \mathbf{w})$.

Let the maximum value of the dual function be $\xi^*$. By the min-max Theorem we have that

$$\xi^* = \max_{\mathbf{w} \in \mathbb{R}^d} \min_{\mathbf{u} \in \mathcal{C}_\rho, \mathbf{v} \in \mathcal{F}} \mathcal{L}(\mathbf{u}, \mathbf{v}, \mathbf{w}) = \min_{\mathbf{u} \in \mathcal{C}_\rho, \mathbf{v} \in \mathcal{F}} \max_{\mathbf{w} \in \mathbb{R}^d} \mathcal{L}(\mathbf{u}, \mathbf{v}, \mathbf{w}) = \min_{\mathbf{u} \in \mathcal{C}_\rho \cap \mathcal{F}} 2\psi(\mathbf{u})$$

The last equality follows from the observation that if $\mathbf{u} \neq \mathbf{v}$ then $\max_{\mathbf{w} \in \mathbb{R}^d} \mathcal{L}(\mathbf{u}, \mathbf{v}, \mathbf{w}) = \infty$.

Let $\mathbf{u}^* \in \mathcal{C}_\rho \cap \mathcal{F}$ such that

$$\psi(\mathbf{u}^*) = \min_{\mathbf{u} \in \mathcal{C}_\rho \cap \mathcal{F}} \psi(\mathbf{u}).$$

Let $\mathcal{W}^* = \operatorname{argmax}_{\mathbf{w} \in \mathbb{R}^d} \xi(\mathbf{w}) \subseteq \mathbb{R}^d$.

**Definition 5** (Primal and Dual gaps). *For any $\mathbf{u} \in \mathcal{C}_\rho, \mathbf{v} \in \mathcal{F}$ and $\mathbf{w} \in \mathbb{R}^d$, we define the primal and dual gaps as follows:*

$$\Delta^{(p)}(\mathbf{u}, \mathbf{v}, \mathbf{w}) = \mathcal{L}(\mathbf{u}, \mathbf{v}, \mathbf{w}) - \min_{\mathbf{u} \in \mathcal{C}_\rho, \mathbf{v} \in \mathcal{F}} \mathcal{L}(\mathbf{u}, \mathbf{v}, \mathbf{w}) = \mathcal{L}(\mathbf{u}, \mathbf{v}, \mathbf{w}) - \xi(\mathbf{w})$$

$$\Delta^{(d)}(\mathbf{w}) = \xi^* - \xi(\mathbf{w}) = 2\psi(\mathbf{u}^*) - \xi(\mathbf{w})$$

*and define the total gap as $\Delta(\mathbf{u}, \mathbf{v}, \mathbf{w}) = \Delta^{(p)}(\mathbf{u}, \mathbf{v}, \mathbf{w}) + \Delta^{(d)}(\mathbf{w})$.*

In the Theorems and Lemmas below, we will refer to the iterates $\mathbf{u}_t, \mathbf{v}_t, \widetilde{\mathbf{u}}_t, \widetilde{\mathbf{v}}_t$ in the Algorithm 1. We use the the short-hands $\Delta_t, \Delta_t^{(p)}, \Delta_t^{(d)}$ for representing the same primal and dual gaps evaluated at, $(\mathbf{u}_{t+1}, \mathbf{v}_{t+1}, \mathbf{w}_t)$.

The overloading of notation for $\rho$ in the definition of the duality gaps and the LMO confusion vector estimation error is intentional. In our analysis of the algorithm using the duality gap, we will set $\rho$ to be exactly equal to the confusion vector estimation error in the `plug-in` algorithm referred to by Algorithm 1.

We will require the use of Theorem 1 and Corollary 1 from Gidel et al. [15], which we restate here in our notation. We use the following facts to transform their Theorem.

$$|\psi(\mathbf{u}) + \psi(\mathbf{v}) - \psi(\mathbf{u}') - \psi(\mathbf{v}')| \leq 2L\|[\mathbf{u} - \mathbf{u}', \mathbf{v} - \mathbf{v}']\|$$
$$\|[I, -I]^\top [-I, I]\| = 2$$
$$(\operatorname{diam}(F))^2 \leq d$$
$$(\operatorname{diam}(\mathcal{C}_\rho))^2 \leq 2d + 2\rho^2$$
$$(\operatorname{diam}(\mathcal{C}_\rho \times \mathcal{F}))^2 \leq 3d + 2\rho^2$$

where $\|M\|$ of a matrix $M$ refers to its spectral norm, and $\operatorname{diam}(\mathcal{A})$ refers to the diameter of a set $\mathcal{A}$, i.e. the maximum $\ell_2$ distance between any two elements from the set $\mathcal{A}$. We will use $\zeta^2$ as a shorthand for $3d + 2\rho^2$.

**Theorem.** *There exists a constant $\alpha > 0$ such that*

$$\xi^* - \xi(\mathbf{w}) \geq \frac{1}{2L_\lambda \zeta^2} \min \left\{ \alpha^2 \operatorname{dist}(\mathbf{w}, \mathcal{W}^*)^2, \alpha L_\lambda \zeta^2 \operatorname{dist}(\mathbf{w}, \mathcal{W}^*) \right\}$$

$$\|\nabla \xi(\mathbf{w})\| \geq \frac{1}{2L_\lambda \zeta^2} \min \left\{ \alpha^2 \operatorname{dist}(\mathbf{w}, \mathcal{W}^*), \alpha L_\lambda \zeta^2 \right\}$$

$$\|\nabla \xi(\mathbf{w})\| \geq \frac{\alpha}{\sqrt{2L_\lambda \zeta^2}} \min \left\{ \sqrt{\xi^* - \xi(\mathbf{w})}, \sqrt{\frac{L_\lambda \zeta^2}{2}} \right\}$$

*where $L_\lambda = 2L + 2\lambda$ and* dist *represents the standard distance function between a point and a set, i.e.* $\operatorname{dist}(\mathbf{x}, \mathcal{A}) = \min_{\mathbf{x}' \in \mathcal{A}} \|\mathbf{x} - \mathbf{x}'\|$.

## A.2 Proof of Proposition 2

**Proposition** (LMO$_\mathcal{C}$ through Bayes-optimal Classifier). *Suppose we wish to minimize $\langle \mathbf{a}, \mathbf{u} \rangle$ over $\mathbf{u} \in \mathcal{C}$. Define the example-dependent loss matrix $\mathbf{L} : \mathcal{X} \to \mathbb{R}^{n \times n}$ as $L_{j,k}(x) = \sum_{i=1}^d a_i \sigma_i(x, j, k)$. Then the solution to*

*the linear minimization problem is directly given by the Bayes-optimal classifier for this loss matrix. Specifically, construct a classifier $g^* : \mathcal{X} \rightarrow [n]$ with*

$$g^*(x) = \operatorname*{argmin}_{\widehat{y} \in [n]} \sum_{j=1}^{n} \eta_j(x) \, L_{j,\widehat{y}}(x),$$

*where $\eta_j(x) = \mathbf{P}(Y = 1|x)$ is the class-conditional probability. Then $C[g^*] \in \operatorname{argmin}_{\mathbf{u} \in \mathcal{C}} \langle \mathbf{a}, \mathbf{u} \rangle$.*

*Proof.*

$$\min_{\mathbf{u} \in \mathcal{C}} \langle \mathbf{a}, \mathbf{u} \rangle = \min_{g \in \mathcal{H}} \sum_{i=1}^{d} a_i \mathbf{E}_{X \sim \mu} \left[ \mathbf{E}_{Y \sim \eta(X)} [\mathbf{E}_{\widehat{Y} \sim g(X)} [\sigma_i(X, Y, \widehat{Y})]] \right]$$

$$= \mathbf{E}_{X \sim \mu} \left[ \min_{\mathbf{g} \in \Delta_n} \mathbf{E}_{\widehat{Y} \sim \mathbf{g}} \left[ \mathbf{E}_{Y \sim \eta(X)} \left[ \sum_{i=1}^{d} a_i \sigma_i(X, Y, \widehat{Y}) \right] \right] \right]$$

$$= \mathbf{E}_{X \sim \mu} \left[ \min_{\widehat{y} \in [n]} \sum_{j=1}^{n} \eta_j(X) \sum_{i=1}^{d} a_i \sigma_i(X, j, \widehat{y}) \right]$$

$$= \mathbf{E}_{X \sim \mu} \left[ \min_{\widehat{y} \in [n]} \sum_{j=1}^{n} \eta_j(X) L_{j,\widehat{y}}(X) \right]$$

Now,

$$\langle \mathbf{a}, C[g^*] \rangle = \sum_{i=1}^{d} a_i \mathbf{E}_{X \sim \mu} \left[ \mathbf{E}_{Y \sim \eta(X)} \left[ \mathbf{E}_{\widehat{Y} \sim g^*(X)} \left[ \sigma_i(X, Y, \widehat{Y}) \right] \right] \right]$$

$$= \mathbf{E}_{X \sim \mu} \left[ \mathbf{E}_{Y \sim \eta(X)} \left[ \sum_{i=1}^{d} a_i \sigma_i(X, Y, g^*(X)) \right] \right]$$

$$= \mathbf{E}_{X \sim \mu} \left[ \sum_{j=1}^{n} \eta_j(X) \sum_{i=1}^{d} a_i \sigma_i(X, j, g^*(X)) \right]$$

$$= \mathbf{E}_{X \sim \mu} \left[ \sum_{j=1}^{n} \eta_j(X) L_{j,g^*(X)}(X) \right]$$

$$= \mathbf{E}_{X \sim \mu} \left[ \min_{\widehat{y} \in [n]} \sum_{j=1}^{n} \eta_j(X) L_{j,\widehat{y}}(X) \right]$$

where the last equation follows from construction of $g^*$. □

## A.3 Proof of Theorem 3

**Theorem.** *Let $h^*$ denote the optimal feasible solution for (OP1), i.e. $\phi_k(C[h^*]) \le 0, \forall k$ and $\psi(C[h^*]) \le \psi(C[h])$ for all $h$ that is feasible. Under the regularity assumptions, for large enough $\lambda$ and an appropriate step-size parameter $C$, there exists an $\bar{\epsilon} > 0$ such that, for all $\epsilon \le \bar{\epsilon}$, and $T \ge \frac{c}{\epsilon^2}$, with probability $1 - \delta$ over draw of the training samples $S$ i.i.d. from $D$, the classifier $\widehat{h}$ returned by Algorithm 1 is near-optimal and near-feasible:*

$$\textit{Optimality}: \quad \psi(C[\widehat{h}]) \le \psi(C[h^*]) + c\sqrt{\omega} + \epsilon,$$
$$\textit{Feasibility}: \quad \phi_k(C[\widehat{h}]) \le c\sqrt{\omega} + L\epsilon, \ \forall k \in [K],$$

*where $\omega = \sqrt{d}\mathbf{E}||\eta(X) - \widehat{\eta}(X)||_1 + d\sqrt{\frac{d \log(d) + \log(Nn^2) + \log(1/\delta)}{N}}$ captures the approximation level of the LMO given by Algorithm 2, and $c > 0$ is a constant not dependent on the number of iterations $T$ and the training samples.*

*Proof.* Firstly, we prove in Corollary 7 that the Algorithm 2 gives an approximate LMO over $\mathcal{C}$ even though it uses only finite data. These Lemmas are more general than those in Narasimhan et al. (2015) [33], because they accommodate more general sufficient statistics functions $\sigma$.

Secondly, we show that the usage of an approximate LMO in Equations (4), and (5) does not affect the convergence results by Gidel et al. [15]. They measure the sub-optimality of an iterate using a duality gap

measure. In Lemma 9 we show that a similar bound on the duality gap can be derived with an approximate LMO over $\mathcal{C}$ as well.

Thirdly, we use the strict feasibilty assumption to convert a bound on the duality gap into a bound on the sub-optimality of problem (OP2) in Lemma 8.

Lemmas 9 can be applied to Lemma 8 setting both $\tau$ and $\kappa$ to be equal to $\frac{c}{T} + c(\rho + \rho')$. In both the inequalities, the $\sqrt{\kappa}$ term dominates, and hence

$$||C[h_b] - \mathbf{v}_b||_2^2 \le c(\rho + \rho') + \frac{c}{T}$$

$$\psi(C[h_b]) \le \min_{\mathbf{u} \in \mathcal{C} \cap \mathcal{F}} \psi(\mathbf{u}) + c\sqrt{\rho + \rho' + \frac{1}{T}}$$

For large enough $T$, these can be simplified as follows,

$$||C[h_b] - \mathbf{v}_b|| \le c\sqrt{\rho + \rho'} + \frac{c}{\sqrt{T}}$$

$$\psi(C[h_b]) \le \min_{\mathbf{u} \in \mathcal{C} \cap \mathcal{F}} \psi(\mathbf{u}) + c\sqrt{\rho + \rho'} + \frac{c}{\sqrt{T}}$$

Observing that $\mathbf{v}_b \in \mathcal{F}$ and the constraint functions $\phi_k$ are $L$-Lipschitz, we get the Theorem statement. The expressions for $\omega = \rho + \rho'$ follow from Corollary 7. $\qquad\square$

### A.3.1   LMO Lemmas

**Lemma 5.** *Let $\mathbf{a} \in \mathbb{R}^d$. Let $\widehat{g}, \widetilde{\mathbf{u}} = \texttt{plug-in}(\mathbf{a})$ as in Algorithm 2, then*

$$\langle \mathbf{a}, C[\widehat{g}] \rangle \le \min_{\mathbf{u} \in \mathcal{C}} \langle \mathbf{a}, \mathbf{u} \rangle + 2||\mathbf{a}||_2 \sqrt{d} \mathbf{E} ||\eta(X) - \widehat{\eta}(X)||_1$$

*for some constant $c_3 > 0$.*

*Proof.* Fix some $\mathbf{a} \in \mathbb{R}^d$. Let $\mathbf{L} : \mathcal{X} \to \mathbb{R}^{n \times n}$, be such that,

$$L_{j,k}(x) = \sum_{i=1}^{d} a_i \sigma_i(x, j, k) \le ||\mathbf{a}||_1 \le \sqrt{d} ||\mathbf{a}||_2$$

From Proposition 2, the Bayes optimal classifier $g^* : \mathcal{X} \to [n]$ is

$$g^*(x) = \operatorname*{argmin}_{\widehat{y} \in [n]} \sum_{j=1}^{n} \eta_j(x) \, L_{j,\widehat{y}}(x),$$

Recall that $\widehat{g}$ is the same as $g^*$ above, with $\eta$ replaced by $\widehat{\eta}$. We have that,

$$\langle \mathbf{a}, C[\widehat{g}] \rangle = \mathbf{E}_X [\mathbf{E}_{Y \sim \eta(X)} [\sum_{i=1}^{d} a_i \sigma_i(X, Y, \widehat{g}(X))]]$$

$$= \mathbf{E}_X \left[ \sum_{y=1}^{n} \eta_y(X) L_{y,\widehat{g}(X)}(X) \right]$$

$$= \mathbf{E}_X \left[ \sum_{y=1}^{n} (\eta_y(X) - \widehat{\eta}_y(X)) L_{y,\widehat{g}(X)}(X) \right] + \mathbf{E}_X \left[ \sum_{y=1}^{n} \widehat{\eta}_y(X) L_{y,\widehat{g}(X)}(X) \right]$$

$$\le ||\mathbf{a}||_2 \sqrt{d} \mathbf{E}_X [||\eta(X) - \widehat{\eta}(X)||_1] + \mathbf{E}_X \left[ \sum_{y=1}^{n} \widehat{\eta}_y(X) L_{y,\widehat{g}(X)}(X) \right]$$

$$\le ||\mathbf{a}||_2 \sqrt{d} \mathbf{E}_X [||\eta(X) - \widehat{\eta}(X)||_1] + \mathbf{E}_X \left[ \sum_{y=1}^{n} \widehat{\eta}_y(X) L_{y,g^*(X)}(X) \right]$$

$$\le ||\mathbf{a}||_2 \sqrt{d} \mathbf{E}_X [||\eta(X) - \widehat{\eta}(X)||_1] + \mathbf{E}_X \left[ \sum_{y=1}^{n} (\widehat{\eta}_y(X) - \eta_y(X) L_{y,g^*(X)}(X) \right] + \mathbf{E}_X \left[ \sum_{y=1}^{n} (\eta_y(X) L_{y,g^*(X)}(X) \right]$$

$$\le 2||\mathbf{a}||_2 \sqrt{d} \mathbf{E}_X [||\eta(X) - \widehat{\eta}(X)||_1] + \min_{\mathbf{u} \in \mathbf{C}} \langle \mathbf{a}, \mathbf{u} \rangle$$

$$\square$$

**Lemma 6.** *Let $\widehat{g}^{\mathbf{a}}, \widetilde{\mathbf{u}}^{\mathbf{a}} = $ plug-in$(\mathbf{a})$ as in Algorithm 2, then with probability $1 - \delta$ over the samples $\{(x_{N/2}, y_{N/2}), \ldots, (x_N, y_N)\}$, we have that for all $\mathbf{a} \in \mathbb{R}^d$*

$$||C[\widehat{g}^{\mathbf{a}}] - \widetilde{\mathbf{u}}^{\mathbf{a}}||_2 \le cd\sqrt{\frac{d \log(d) + \log(Nn^2) + \log(1/\delta)}{N}}$$

*where $c$ is an absolute constant.*

*Proof.* Fix some $z \in [d]$. We have for any $\mathbf{a} \in \mathbb{R}^d$,

$$C_z[\widehat{g}^{\mathbf{a}}] = \mathbf{E}_{X,Y}\sigma_z(X, Y, \widehat{g}^{\mathbf{a}}(X)) = \mathbf{E}_D\left[\sigma_z(X, Y, \widehat{g}^{\mathbf{a}}(X))\right]$$

$$\widetilde{u}_z^{\mathbf{a}} = \frac{2}{N}\sum_{j=N/2}^N \sigma_z(x_j, y_j, \widehat{g}^{\mathbf{a}}(x_j)) = \mathbf{E}_S\left[\sigma_z(X, Y, \widehat{g}^{\mathbf{a}}(X))\right]$$

where we abuse notation by denoting the empirical expectation over the last $N/2$ samples as $\mathbf{E}_S$ and the population expectation as $\mathbf{E}_D$.

For any $x \in \mathcal{X}, \widehat{y} \in [n]$, let $\theta(x, \widehat{y}) \in \mathbb{R}^d$ be such that

$$\theta_i(x, \widehat{y}) = \sum_{y=1}^n \widehat{\eta}_y(x)\sigma_i(x, y, \widehat{y}).$$

Then, by definition of $\widehat{g}^{\mathbf{a}}$, we have that

$$\widehat{g}^{\mathbf{a}}(x) = \text{argmin}_{\widehat{y} \in [n]}\, \mathbf{a}^\top \boldsymbol{\theta}(x, \widehat{y}).$$

We have that the Natarajan dimension $d_{\text{Nat}}$ of the function class

$$\mathcal{G} = \{\widehat{g}^{\mathbf{a}}(x) = \text{argmin}_{\widehat{y} \in [n]}\, \mathbf{a}^\top \theta(x, \widehat{y}) : \mathbf{a} \in \mathbb{R}^d\} \subseteq [n]^{\mathcal{X}}$$

is $O(d \log(d))$ [36]. The growth function $\Pi_{\mathcal{G}}(N)$ denoting the number of distinct labellings of $N$ points is given by Cesa-Bianchi and Haussler [6] as,

$$\Pi_{\mathcal{G}}(N) \le d_{\text{Nat}}N^{d_{\text{Nat}}}n^{2d_{\text{Nat}}}.$$

Using standard Hoeffding inequality and uniform convergence arguments, we have that with probability $1 - \delta$

$$\sup_{\mathbf{a} \in \mathbb{R}^d} |C_z[\widehat{g}^{\mathbf{a}}] - \widetilde{u}_z^{\mathbf{a}}| = \sup_{g \in \mathcal{G}} |\mathbf{E}_S\left[\sigma_z(X, Y, g(X))\right] - \mathbf{E}_D\left[\sigma_z(X, Y, g(X))\right]|$$

$$\le c\left(\sqrt{\frac{\log(\Pi_{\mathcal{G}}(N)) + \log(1/\delta)}{N}}\right)$$

$$\le c\left(\sqrt{\frac{d\log(d) + \log(Nn^2) + \log(1/\delta)}{N}}\right)$$

We thus have that,

$$\sup_{\mathbf{a} \in \mathbb{R}^d} ||C[\widehat{g}^{\mathbf{a}}] - \widetilde{\mathbf{u}}^{\mathbf{a}}||_2 \le \sup_{\mathbf{a} \in \mathbb{R}^d} ||C[\widehat{g}^{\mathbf{a}}] - \widetilde{\mathbf{u}}^{\mathbf{a}}||_1$$

$$= \sup_{\mathbf{a} \in \mathbb{R}^d} \sum_{z=1}^d |C_z[\widehat{g}^{\mathbf{a}}] - \widetilde{u}_z^{\mathbf{a}}|$$

$$\le \sum_{z=1}^d \sup_{\mathbf{a} \in \mathbb{R}^d} |C_z[\widehat{g}^{\mathbf{a}}] - \widetilde{u}_z^{\mathbf{a}}|$$

$$\le c\left(d\sqrt{\frac{d\log(d) + \log(Nn^2) + \log(1/\delta)}{N}}\right).$$

where the last statement holds with probability $1 - \delta$. $\square$

**Corollary 7.** *The function* plug-in *in Algorithm 2 is a $(\rho, \rho', \delta)$-approximate LMO with $\rho = cd\sqrt{\frac{d\log(d) + \log(Nn^2) + \log(1/\delta)}{N}}$ and $\rho' = 2\sqrt{d}\mathbf{E}||\eta(X) - \widehat{\eta}(X)||_1$ for some constant $c > 0$.*

We will fix a $\delta$ probability of failure throughout the rest of the proof, and assume that the training sample $S$ is "good", in which case the empirical confusion vector output by the plug-in algorithm is $\rho$ close to the true confusion vector of the classifier whenever it is called by Algorithm 1.

### A.3.2 Converting Duality Gap Bounds to Primal Sub-Optimality Bounds

**Lemma 8.** *Let* $\mathbf{g} : \mathcal{X} \to \Delta_n$ *be a randomized classifier, and* $\mathbf{u} \in \mathbb{R}^d$ *be such that* $\|\mathbf{u} - C[\mathbf{g}]\| \leq \rho$. *Let* $\mathbf{v} \in \mathcal{F}, \mathbf{w} \in \mathbb{R}^d$ *be such that* $\Delta(\mathbf{u}, \mathbf{v}, \mathbf{w}) \leq \tau$ *and* $\|\mathbf{u} - \mathbf{v}\|^2 \leq \kappa$. *Then,*

$$\psi(C[\mathbf{g}]) \leq \min_{\mathbf{u}' \in \mathcal{C} \cap \mathcal{F}} \psi(\mathbf{u}') + c\tau + c\sqrt{\kappa} + L\rho$$

$$\|C[\mathbf{g}] - \mathbf{v}\| \leq \rho + \sqrt{\kappa}$$

*for some constant* $c > 0$.

*Proof.* The second inequality in the lemma trivially follows from the triangle inequality. We will prove the first inequality below.

By construction, $\mathbf{u} \in \mathcal{C}_\rho$. As $\Delta(\mathbf{u}, \mathbf{v}, \mathbf{w}) \leq \tau$ we have,

$$\Delta^{(\mathrm{p})}(\mathbf{u}, \mathbf{v}, \mathbf{w}) = \mathcal{L}(\mathbf{u}, \mathbf{v}, \mathbf{w}) - \min_{\mathbf{u}' \in \mathcal{C}_\rho, \mathbf{v}' \in \mathcal{F}} \mathcal{L}(\mathbf{u}', \mathbf{v}', \mathbf{w}) \leq \tau \tag{6}$$

$$\Delta^{(\mathrm{d})}(\mathbf{w}) = 2\psi(\mathbf{u}^*) - \min_{\mathbf{u}' \in \mathcal{C}_\rho, \mathbf{v}' \in \mathcal{F}} \mathcal{L}(\mathbf{u}', \mathbf{v}', \mathbf{w}) \leq \tau \tag{7}$$

where $\mathbf{u}^* \in \mathrm{argmin}_{\mathbf{u}' \in \mathcal{C}_\rho \cap \mathcal{F}} \psi(\mathbf{u}')$. Setting $\mathbf{u}' = \mathbf{v}' = \mathbf{u}^*$ in the second term of Eqn. (6), we get

$$\psi(\mathbf{u}) + \psi(\mathbf{v}) + \mathbf{w}^T(\mathbf{u} - \mathbf{v}) + \frac{\lambda}{2}\|\mathbf{u} - \mathbf{v}\|^2 \leq 2\psi(\mathbf{u}^*) + \tau. \tag{8}$$

Now from our assumption that there exists a ball of radius $r$ contained in $\mathcal{C} \cap \mathcal{F}$, we can set $\mathbf{u}', \mathbf{v}' = \mathbf{c} \pm \frac{r}{\|\mathbf{w}\|}\mathbf{w}$ in the second term of Eqn. (7) to get

$$2\psi(\mathbf{u}^*) \leq \psi(\mathbf{u}') + \psi(\mathbf{v}') - 2r\|\mathbf{w}\| + 2\lambda r^2 + \tau. \tag{9}$$

This can be reduced to a bound on $\|\widehat{\mathbf{w}}\|$,

$$\|\mathbf{w}\| \leq \frac{2R}{r} + 2\lambda r + \frac{\tau}{r} \tag{10}$$

Eqn. (8) becomes the following by Cauchy-Schwarz:

$$\psi(\mathbf{u}) + \psi(\mathbf{v}) \leq 2\psi(\mathbf{u}^*) + \tau - \mathbf{w}^\top(\mathbf{u} - \mathbf{v}) - \frac{\lambda}{2}\|\mathbf{u} - \mathbf{v}\|^2 \leq 2\psi(\mathbf{u}^*) + \tau + \left(\frac{2R}{r} + 2\lambda r + \frac{\tau}{r}\right)\sqrt{\kappa}. \tag{11}$$

As $\psi$ is $L$-Lipschitz, we have

$$\psi(\mathbf{u}) - \psi(\mathbf{v}) \leq L\|\mathbf{u} - \mathbf{v}\| \leq L\sqrt{\kappa} \tag{12}$$

Adding Eqns. (11) and 12 and dividing by 2, we get

$$\psi(\mathbf{u}) \leq \min_{\mathbf{u}' \in \mathcal{C}_\rho \cap \mathcal{F}} \psi(\mathbf{u}') + \frac{\tau}{2} + \frac{\left(\frac{2R}{r} + 2\lambda r + \frac{\tau}{r}\right) + L}{2}\sqrt{\kappa}$$

As $\mathcal{C}_\rho \supseteq \mathcal{C}$, and $\psi$ is $L$-Lipschitz, we have

$$\psi(C[\mathbf{g}]) \leq \psi(\mathbf{u}) + L\|\mathbf{u} - C[\mathbf{g}]\|$$

$$\leq \min_{\mathbf{u}' \in \mathcal{C}_\rho \cap \mathcal{F}} \psi(\mathbf{u}') + \frac{\tau}{2} + \frac{\left(\frac{2R}{r} + 2\lambda r + \frac{\tau}{r}\right) + L}{2}\sqrt{\kappa} + L\rho$$

$$\leq \min_{\mathbf{u}' \in \mathcal{C} \cap \mathcal{F}} \psi(\mathbf{u}') + \frac{\tau}{2} + \frac{\left(\frac{2R}{r} + 2\lambda r + \frac{\tau}{r}\right) + L}{2}\sqrt{\kappa} + L\rho$$

$\square$

### A.3.3 Bounding the Duality Gap

**Lemma 9.** *Let* $b \in [T]$ *be such that* $\widehat{h} = h_b$ *in Algorithm 1. Let the* `plug-in` *sub-routine used be a* $(\rho, \rho', \delta)$-*approximate LMO. For large enough* $T$ *and* $\lambda$, *with probability* $1 - \delta$ *over the training samples we have that*

$$\Delta(\mathbf{u}_b, \mathbf{v}_b, \mathbf{w}_{b-1}) \leq c(\rho + \rho') + \frac{c}{T}$$

$$\|\mathbf{u}_b - \mathbf{v}_b\|^2 \leq c(\rho + \rho') + \frac{c}{T}$$

*where* $h_b, \mathbf{v}_b, \mathbf{w}_{b-1}$ *are as defined in Algorithm 1, and* $c$ *is a constant independent of* $\rho, \rho'$ *and* $T$.

*Proof.* For large $\lambda$ and $T$, the conditions in Corollary 16 and Lemma 17 are satisfied, and hence the Lemma follows directly. □

**Lemma 10.** *For all $\mathbf{u} \in \mathcal{C}_\rho$, $\mathbf{v} \in \mathcal{F}$ and $\mathbf{w} \in \mathbb{R}^d$*

$$\|(\mathbf{u} - \mathbf{v}) - (\widehat{\mathbf{u}}(\mathbf{w}) - \widehat{\mathbf{v}}(\mathbf{w}))\|^2 \leq \frac{2}{\lambda} \left( \mathcal{L}(\mathbf{u}, \mathbf{v}, \mathbf{w}) - \mathcal{L}(\widehat{\mathbf{u}}(\mathbf{w}), \widehat{\mathbf{v}}(\mathbf{w}), \mathbf{w}) \right) \tag{13}$$

*where $\widehat{\mathbf{u}}(\mathbf{w}), \widehat{\mathbf{v}}(\mathbf{w}) \in \mathrm{argmin}_{\mathbf{u} \in \mathcal{C}_\rho, \mathbf{v} \in \mathcal{F}} \mathcal{L}(\mathbf{u}, \mathbf{v}, \mathbf{w})$ are functions of $\mathbf{w}$.*

*Proof.* We drop the dependence on $\mathbf{w}$ in $\widehat{\mathbf{u}}, \widehat{\mathbf{v}}$ for simplicity below.

By convexity of $\psi$ we have that,

$$\psi(\mathbf{u}) - \psi(\widehat{\mathbf{u}}) \geq (\nabla\psi(\widehat{\mathbf{u}}))^T (\mathbf{u} - \widehat{\mathbf{u}}) \text{ and } \psi(\mathbf{v}) - \psi(\widehat{\mathbf{v}}) \geq (\nabla\psi(\widehat{\mathbf{v}}))^T (\mathbf{v} - \widehat{\mathbf{v}})$$

then by simple algebra,

$$\mathcal{L}(\mathbf{u}, \mathbf{v}, \mathbf{w}) - \mathcal{L}(\widehat{\mathbf{u}}, \widehat{\mathbf{v}}, \mathbf{w})$$

$$= \psi(\mathbf{u}) - \psi(\widehat{\mathbf{u}}) + \psi(\mathbf{v}) - \psi(\widehat{\mathbf{v}}) + \mathbf{w}^\top (\mathbf{u} - \mathbf{v} - \widehat{\mathbf{u}} + \widehat{\mathbf{v}}) + \frac{\lambda}{2}(\|\mathbf{u} - \mathbf{v}\|^2 - \|\widehat{\mathbf{u}} - \widehat{\mathbf{v}}\|^2)$$

$$\geq (\nabla\psi(\widehat{\mathbf{u}}) + \mathbf{w})^\top (\mathbf{u} - \widehat{\mathbf{u}}) + (\nabla\psi(\widehat{\mathbf{v}}) - \mathbf{w})^\top (\mathbf{v} - \widehat{\mathbf{v}}) + \frac{\lambda}{2}(\|\mathbf{u} - \mathbf{v}\|^2 - \|\widehat{\mathbf{u}} - \widehat{\mathbf{v}}\|^2)$$

$$= (\nabla\psi(\widehat{\mathbf{u}}) + \mathbf{w} + \lambda(\widehat{\mathbf{u}} - \widehat{\mathbf{v}}))^\top (\mathbf{u} - \widehat{\mathbf{u}}) + (\nabla\psi(\widehat{\mathbf{v}}) - \mathbf{w} - \lambda(\widehat{\mathbf{u}} - \widehat{\mathbf{v}}))^\top (\mathbf{v} - \widehat{\mathbf{v}})$$

$$\quad + \frac{\lambda}{2}\|(\mathbf{u} - \mathbf{v}) - (\widehat{\mathbf{u}} - \widehat{\mathbf{v}})\|^2$$

$$= (\nabla_{\mathbf{u}}\mathcal{L}(\widehat{\mathbf{u}}, \widehat{\mathbf{v}}, \mathbf{w}))^\top (\mathbf{u} - \widehat{\mathbf{u}}) + (\nabla_{\mathbf{v}}\mathcal{L}(\widehat{\mathbf{u}}, \widehat{\mathbf{v}}, \mathbf{w}))^\top (\mathbf{v} - \widehat{\mathbf{v}}) + \frac{\lambda}{2}\|(\mathbf{u} - \mathbf{v}) - (\widehat{\mathbf{u}} - \widehat{\mathbf{v}})\|^2$$

$$\geq \frac{\lambda}{2}\|(\mathbf{u} - \mathbf{v}) - (\widehat{\mathbf{u}} - \widehat{\mathbf{v}})\|^2$$

The last inequality follows from the definition $\widehat{\mathbf{u}}, \widehat{\mathbf{v}}$. □

The next lemma captures the essence of what happens in one iteration of Algorithm 1 in lines 5-8. We use the same symbols as in the algorithm for ease of reference.

**Lemma 11.** *Let $\mathbf{u}_{t-1} \in \mathcal{C}_\rho, \mathbf{v}_{t-1} \in \mathcal{F}, \mathbf{w}_{t-1} \in \mathbb{R}^d$. Let $\mathbf{a}_{t-1} = \nabla_{\mathbf{u}}\mathcal{L}(\mathbf{u}_{t-1}, \mathbf{v}_{t-1}, \mathbf{w}_{t-1})$ and $\mathbf{b}_{t-1} = \nabla_{\mathbf{v}}\mathcal{L}(\mathbf{u}_{t-1}, \mathbf{v}_{t-1}, \mathbf{w}_{t-1})$. Let $\Omega$ be a $(\rho, \rho', \delta)$-approximate LMO. Let $\widehat{g}_t, \widetilde{\mathbf{u}}_t = \Omega(\mathbf{a}_{t-1}; S)$, and $\widetilde{\mathbf{v}}_t \in \mathrm{argmin}_{\mathbf{v} \in \mathcal{F}}\langle \mathbf{b}_{t-1}, \mathbf{v} \rangle$. Let $\mathbf{u}_t = (1 - \gamma_t)\mathbf{u}_{t-1} + \gamma_t\widetilde{\mathbf{u}}_t$ and $\mathbf{v}_t = (1 - \gamma_t)\mathbf{v}_{t-1} + \gamma_t\widetilde{\mathbf{v}}_t$. Let $\widehat{\mathbf{u}}_{t-1}, \widehat{\mathbf{v}}_{t-1} = \widehat{\mathbf{u}}(\mathbf{w}_{t-1}), \widehat{\mathbf{v}}(\mathbf{w}_{t-1})$ as defined in Lemma 10. Then*

$$\mathcal{L}(\mathbf{u}_t, \mathbf{v}_t, \mathbf{w}_{t-1}) - \mathcal{L}(\widehat{\mathbf{u}}_{t-1}, \widehat{\mathbf{v}}_{t-1}, \mathbf{w}_{t-1})$$

$$\leq (1 - \gamma_t) \left( \mathcal{L}(\mathbf{u}_{t-1}, \mathbf{v}_{t-1}, \mathbf{w}_{t-1}) - \mathcal{L}(\widehat{\mathbf{u}}_{t-1}, \widehat{\mathbf{v}}_{t-1}, \mathbf{w}_{t-1}) \right) + \gamma_t\|\mathbf{a}_{t-1}\|(\rho + \rho') + \frac{1}{2}\beta_\lambda\gamma_t^2\zeta^2$$

*Proof.* Using smoothness,

$$\mathcal{L}(\mathbf{u}_t, \mathbf{v}_t, \mathbf{w}_{t-1}) - \mathcal{L}(\mathbf{u}_{t-1}, \mathbf{v}_{t-1}, \mathbf{w}_{t-1})$$

$$\leq \nabla_{\mathbf{u}}\mathcal{L}(\mathbf{u}_{t-1}, \mathbf{v}_{t-1}, \mathbf{w}_{t-1})^\top[\mathbf{u}_t - \mathbf{u}_{t-1}] + \nabla_{\mathbf{v}}\mathcal{L}(\mathbf{u}_{t-1}, \mathbf{v}_{t-1}, \mathbf{w}_{t-1})^\top[\mathbf{v}_t - \mathbf{v}_{t-1}] + \frac{\beta_\lambda}{2}\left( \|\mathbf{u}_t - \mathbf{u}_{t-1}\|^2 + \|\mathbf{v}_t - \mathbf{v}_{t-1}\|^2 \right)$$

$$= \mathbf{a}_{t-1}^\top[\gamma_t(\widetilde{\mathbf{u}}_t - \mathbf{u}_{t-1})] + \mathbf{b}_{t-1}^\top[\gamma_t(\widetilde{\mathbf{v}}_t - \mathbf{v}_{t-1})] + \frac{\beta_\lambda}{2}\gamma_t^2\left( \|\widetilde{\mathbf{u}}_t - \mathbf{u}_{t-1}\|^2 \right) + \frac{\beta_\lambda}{2}\gamma^2\left( \|\widetilde{\mathbf{v}}_t - \mathbf{v}_{t-1}\|^2 \right)$$

$$\leq \gamma_t\mathbf{a}_{t-1}^\top(\widetilde{\mathbf{u}}_t - \mathbf{u}_{t-1}) + \gamma_t\mathbf{b}_{t-1}^\top(\widetilde{\mathbf{v}}_t - \mathbf{v}_{t-1}) + \gamma_t^2\frac{\beta_\lambda}{2}(\mathrm{diam}^2(\mathcal{C}_\rho) + \mathrm{diam}^2(\mathcal{F}))$$

$$\leq \gamma_t\mathbf{a}_{t-1}^\top(\widetilde{\mathbf{u}}_t - \mathbf{u}_{t-1}) + \gamma_t\mathbf{b}_{t-1}^\top(\widehat{\mathbf{v}}_{t-1} - \mathbf{v}_{t-1}) + \frac{1}{2}\beta_\lambda\gamma_t^2\zeta^2$$

$$\leq \gamma_t\mathbf{a}_{t-1}^\top(\widehat{\mathbf{u}}_{t-1} - \mathbf{u}_{t-1}) + \gamma_t\|\mathbf{a}_{t-1}\|\rho' + \gamma_t\|\mathbf{a}_{t-1}\|\rho + \gamma_t\mathbf{b}_{t-1}^\top(\widetilde{\mathbf{v}}_t - \mathbf{v}_{t-1}) + \frac{1}{2}\beta_\lambda\gamma_t^2\zeta^2$$

$$\leq \gamma_t \left( \mathcal{L}(\widehat{\mathbf{u}}_{t-1}, \widehat{\mathbf{v}}_{t-1}, \mathbf{w}_{t-1}) - \mathcal{L}(\mathbf{u}_{t-1}, \mathbf{v}_{t-1}, \mathbf{w}_{t-1}) \right) + \gamma_t\|\mathbf{a}_{t-1}\|(\rho + \rho') + \frac{1}{2}\beta_\lambda\gamma_t^2\zeta^2$$

Rearranging the terms we get the statement of the lemma. □

The next lemma captures the essence of what happens in one iteration of Algorithm 1 in Line 9. We use the same symbols as in the algorithm for ease of reference.

**Lemma 12** (Variant of Fundamental Descent Lemma in Gidel et al. (2018) [15]). *Let* $\mathbf{w}_t \in \mathbb{R}^d, \mathbf{u}_{t+1} \in \mathcal{C}_\rho, \mathbf{v}_{t+1} \in \mathcal{F}$. *Let* $\mathbf{w}_{t+1} = \mathbf{w}_t + \eta_t(\mathbf{u}_{t+1} - \mathbf{v}_{t+1})$. *Then,*

$$\Delta_{t+1} - \Delta_t \leq \mathcal{L}(\mathbf{u}_{t+2}, \mathbf{v}_{t+2}, \mathbf{w}_{t+1}) - \mathcal{L}(\mathbf{u}_{t+1}, \mathbf{v}_{t+1}, \mathbf{w}_{t+1}) + \frac{2\eta_t}{\lambda}\left(\mathcal{L}(\mathbf{u}_{t+1}, \mathbf{v}_{t+1}, \mathbf{w}_{t+1}) - \mathcal{L}(\widehat{\mathbf{u}}_{t+1}, \widehat{\mathbf{v}}_{t+1}, \mathbf{w}_{t+1})\right)$$
$$- \frac{\eta_t \alpha^2}{2L_\lambda \zeta^2} \min\left\{\Delta_{t+1}^{(d)}, \frac{L_\lambda \zeta^2}{2}\right\}$$

*where* $\alpha > 0$ *is as defined in Section A.1.2. (Also Theorem 1 of Gidel et al. (2018) [15])*

*Proof.* Let $\widehat{\mathbf{u}}_t, \widehat{\mathbf{v}}_t \in \operatorname{argmin}_{\mathbf{u} \in \mathcal{C}_\rho, \mathbf{v} \in \mathcal{F}} \mathcal{L}(\mathbf{u}, \mathbf{v}, \mathbf{w}_t)$. We have that,

$$\begin{aligned}
\Delta_{t+1}^{(d)} - \Delta_t^{(d)} &= \xi(\mathbf{w}_t) - \xi(\mathbf{w}_{t+1}) \\
&= \mathcal{L}(\widehat{\mathbf{u}}_t, \widehat{\mathbf{v}}_t, \mathbf{w}_t) - \mathcal{L}(\widehat{\mathbf{u}}_{t+1}, \widehat{\mathbf{v}}_{t+1}, \mathbf{w}_{t+1}) \\
&\leq \mathcal{L}(\widehat{\mathbf{u}}_{t+1}, \widehat{\mathbf{v}}_{t+1}, \mathbf{w}_t) - \mathcal{L}(\widehat{\mathbf{u}}_{t+1}, \widehat{\mathbf{v}}_{t+1}, \mathbf{w}_{t+1}) \\
&= \langle \mathbf{w}_t - \mathbf{w}_{t+1}, \widehat{\mathbf{u}}_{t+1} - \widehat{\mathbf{v}}_{t+1} \rangle \\
&= -\eta_t \langle \mathbf{u}_{t+1} - \mathbf{v}_{t+1}, \widehat{\mathbf{u}}_{t+1} - \widehat{\mathbf{v}}_{t+1} \rangle
\end{aligned}$$

$$\begin{aligned}
\Delta_{t+1}^{(p)} - \Delta_t^{(p)} &= \Delta^{(p)}(\mathbf{u}_{t+2}, \mathbf{v}_{t+2}, \mathbf{w}_{t+1}) - \Delta^{(p)}(\mathbf{u}_{t+1}, \mathbf{v}_{t+1}, \mathbf{w}_t) \\
&= \mathcal{L}(\mathbf{u}_{t+2}, \mathbf{v}_{t+2}, \mathbf{w}_{t+1}) - \mathcal{L}(\mathbf{u}_{t+1}, \mathbf{v}_{t+1}, \mathbf{w}_t) + \xi(\mathbf{w}_t) - \xi(\mathbf{w}_{t+1}) \\
&= \mathcal{L}(\mathbf{u}_{t+2}, \mathbf{v}_{t+2}, \mathbf{w}_{t+1}) - \mathcal{L}(\mathbf{u}_{t+1}, \mathbf{v}_{t+1}, \mathbf{w}_{t+1}) + \langle \mathbf{w}_{t+1} - \mathbf{w}_t, \mathbf{u}_{t+1} - \mathbf{v}_{t+1} \rangle + \xi(\mathbf{w}_t) - \xi(\mathbf{w}_{t+1}) \\
&= \mathcal{L}(\mathbf{u}_{t+2}, \mathbf{v}_{t+2}, \mathbf{w}_{t+1}) - \mathcal{L}(\mathbf{u}_{t+1}, \mathbf{v}_{t+1}, \mathbf{w}_{t+1}) + \eta_t \|\mathbf{u}_{t+1} - \mathbf{v}_{t+1}\|^2 + \xi(\mathbf{w}_t) - \xi(\mathbf{w}_{t+1})
\end{aligned}$$

Putting both the bounds together, we get,

$$\begin{aligned}
&\Delta_{t+1} - \Delta_t \\
&\leq \mathcal{L}(\mathbf{u}_{t+2}, \mathbf{v}_{t+2}, \mathbf{w}_{t+1}) - \mathcal{L}(\mathbf{u}_{t+1}, \mathbf{v}_{t+1}, \mathbf{w}_{t+1}) + \eta_t \|\mathbf{u}_{t+1} - \mathbf{v}_{t+1}\|^2 - 2\eta_t \langle \mathbf{u}_{t+1} - \mathbf{v}_{t+1}, \widehat{\mathbf{u}}_{t+1} - \widehat{\mathbf{v}}_{t+1} \rangle \\
&= \mathcal{L}(\mathbf{u}_{t+2}, \mathbf{v}_{t+2}, \mathbf{w}_{t+1}) - \mathcal{L}(\mathbf{u}_{t+1}, \mathbf{v}_{t+1}, \mathbf{w}_{t+1}) + \eta_t \|(\mathbf{u}_{t+1} - \mathbf{v}_{t+1}) - (\widehat{\mathbf{u}}_{t+1} - \widehat{\mathbf{v}}_{t+1})\|^2 - \eta_t \|\widehat{\mathbf{u}}_{t+1} - \widehat{\mathbf{v}}_{t+1}\|^2 \\
&\leq \mathcal{L}(\mathbf{u}_{t+2}, \mathbf{v}_{t+2}, \mathbf{w}_{t+1}) - \mathcal{L}(\mathbf{u}_{t+1}, \mathbf{v}_{t+1}, \mathbf{w}_{t+1}) + \frac{2\eta_t}{\lambda}\left(\mathcal{L}(\mathbf{u}_{t+1}, \mathbf{v}_{t+1}, \mathbf{w}_{t+1}) - \mathcal{L}(\widehat{\mathbf{u}}_{t+1}, \widehat{\mathbf{v}}_{t+1}, \mathbf{w}_{t+1})\right) \\
&\quad - \eta_t \|\widehat{\mathbf{u}}_{t+1} - \widehat{\mathbf{v}}_{t+1}\|^2 \tag{14}
\end{aligned}$$

The last inequality above follows from Lemma 10.

We have that $\widehat{\mathbf{u}}_{t+1} - \widehat{\mathbf{v}}_{t+1} = \nabla \xi(\mathbf{w}_{t+1})$, and by Theorem 1 of Gidel et al. [15] (also in Section A.1.2), we have that

$$\|\nabla \xi(\mathbf{w}_{t+1})\|^2 \geq \frac{\alpha^2}{2L_\lambda \zeta^2} \min\left\{\Delta_{t+1}^{(d)}, \frac{L_\lambda \zeta^2}{2}\right\}.$$

Putting it together we get

$$\Delta_{t+1} - \Delta_t \leq \mathcal{L}(\mathbf{u}_{t+2}, \mathbf{v}_{t+2}, \mathbf{w}_{t+1}) - \mathcal{L}(\mathbf{u}_{t+1}, \mathbf{v}_{t+1}, \mathbf{w}_{t+1}) + \frac{2\eta_t}{\lambda}\left(\mathcal{L}(\mathbf{u}_{t+1}, \mathbf{v}_{t+1}, \mathbf{w}_{t+1}) - \mathcal{L}(\widehat{\mathbf{u}}_{t+1}, \widehat{\mathbf{v}}_{t+1}, \mathbf{w}_{t+1})\right)$$
$$- \frac{\eta_t \alpha^2}{2L_\lambda \zeta^2} \min\left\{\Delta_{t+1}^{(d)}, \frac{L_\lambda \zeta^2}{2}\right\}$$

$\square$

We use Lemma 13 to prove Lemma 9. The proof of Lemma 13 closely follows the proof of Theorem 2 in [15] and is split into Lemmas 14, 15, 17. However, we make the iterates $C[h_t]$, $\mathbf{v}_t$ over the set $\mathcal{C}$ and $\mathcal{F}$ explicit and derive results taking into account the approximate LMO for the set $\mathcal{C}$.

**Lemma 13.**
$$\Delta_{t+1} - \Delta_t \leq -\frac{2}{t+2}\min(\Delta_{t+1}, \theta_1) + \frac{\theta_2}{(t+2)^2} + \frac{\theta_3}{t+2}(\rho + \rho')$$

*where* $\theta_1 = \frac{L_\lambda \zeta^2}{2}$, $\theta_2 = 32\frac{\beta_\lambda \zeta^2}{\chi^2 \lambda^2}\left(1 + \frac{2}{\chi\lambda}\right)$ *and* $\theta_3 = \frac{8}{\chi\lambda}\left(1 + \frac{2}{\chi\lambda}\right)\max_t \|\mathbf{a}_{t+1}\|$.

*Proof.* Lemma 12 and Lemma 11 leads to the following equation holding for $\gamma \in [0, 1]$,

$$\Delta_{t+1} - \Delta_t \leq \mathcal{L}(\mathbf{u}_{t+2}, \mathbf{v}_{t+2}, \mathbf{w}_{t+1}) - \mathcal{L}(\mathbf{u}_{t+1}, \mathbf{v}_{t+1}, \mathbf{w}_{t+1}) + \frac{2\eta_t}{\lambda} \left( \mathcal{L}(\mathbf{u}_{t+1}, \mathbf{v}_{t+1}, \mathbf{w}_{t+1}) - \mathcal{L}(\widehat{\mathbf{u}}_{t+1}, \widehat{\mathbf{v}}_{t+1}, \mathbf{w}_{t+1}) \right)$$

$$- \frac{\eta_t \alpha^2}{2L_\lambda \zeta^2} \min \left\{ \Delta_{t+1}^{(\mathrm{d})}, \frac{L_\lambda \zeta^2}{2} \right\}$$

$$\leq \gamma_{t+2} \left( \mathcal{L}(\widehat{\mathbf{u}}_{t+1}, \widehat{\mathbf{v}}_{t+1}, \mathbf{w}_{t+1}) - \mathcal{L}(\mathbf{u}_{t+1}, \mathbf{v}_{t+1}, \mathbf{w}_{t+1}) \right) + \gamma_{t+2} \|\mathbf{a}_{t+1}\| (\rho + \rho') + \frac{1}{2} \beta_\lambda \gamma_{t+2}^2 \zeta^2$$

$$+ \frac{2\eta_t}{\lambda} \left( \mathcal{L}(\mathbf{u}_{t+1}, \mathbf{v}_{t+1}, \mathbf{w}_{t+1}) - \mathcal{L}(\widehat{\mathbf{u}}_{t+1}, \widehat{\mathbf{v}}_{t+1}, \mathbf{w}_{t+1}) \right) - \frac{\eta_t \alpha^2}{2L_\lambda \zeta^2} \min \left\{ \Delta_{t+1}^{(\mathrm{d})}, \frac{L_\lambda \zeta^2}{2} \right\}$$

$$= \left( \frac{2\eta_t}{\lambda} - \gamma_{t+2} \right) \left( \mathcal{L}(\mathbf{u}_{t+1}, \mathbf{v}_{t+1}, \mathbf{w}_{t+1}) - \mathcal{L}(\widehat{\mathbf{u}}_{t+1}, \widehat{\mathbf{v}}_{t+1}, \mathbf{w}_{t+1}) \right) + \gamma_{t+2} \|\mathbf{a}_{t+1}\| (\rho + \rho') + \frac{1}{2} \beta_\lambda \gamma_{t+2}^2 \zeta^2$$

$$- \frac{\eta_t \alpha^2}{2L_\lambda \zeta^2} \min \left\{ \Delta_{t+1}^{(\mathrm{d})}, \frac{L_\lambda \zeta^2}{2} \right\}$$

Then for $\gamma_{t+2} = \frac{4\eta_t}{\lambda}$ we get,

$$\Delta_{t+1} - \Delta_t \leq - \left( \frac{2\eta_t}{\lambda} \right) \left( \mathcal{L}(\mathbf{u}_{t+1}, \mathbf{v}_{t+1}, \mathbf{w}_{t+1}) - \mathcal{L}(\widehat{\mathbf{u}}_{t+1}, \widehat{\mathbf{v}}_{t+1}, \mathbf{w}_{t+1}) \right) + \frac{4\eta_t}{\lambda} \|\mathbf{a}_{t+1}\| (\rho + \rho') + \beta_\lambda \frac{8\eta_t^2}{\lambda^2} \zeta^2$$

$$- \frac{\eta_t \alpha^2}{2L_\lambda \zeta^2} \min \left\{ \Delta_{t+1}^{(\mathrm{d})}, \frac{L_\lambda \zeta^2}{2} \right\} \tag{15}$$

We also have that,

$$\mathcal{L}(\mathbf{u}_{t+2}, \mathbf{v}_{t+2}, \mathbf{w}_{t+1}) \leq \mathcal{L}(\mathbf{u}_{t+1}, \mathbf{v}_{t+1}, \mathbf{w}_{t+1}) + \langle \mathbf{a}_{t+1}, \mathbf{u}_{t+2} - \mathbf{u}_{t+1} \rangle + \langle \mathbf{b}_{t+1}, \mathbf{v}_{t+2} - \mathbf{v}_{t+1} \rangle$$

$$+ \frac{\beta_\lambda}{2} (\|\mathbf{u}_{t+2} - \mathbf{u}_{t+1}\|^2 + \|\mathbf{v}_{t+2} - \mathbf{v}_{t+1}\|^2)$$

$$= \mathcal{L}(\mathbf{u}_{t+1}, \mathbf{v}_{t+1}, \mathbf{w}_{t+1}) + \gamma_{t+2} \langle \mathbf{a}_{t+1}, \widetilde{\mathbf{u}}_{t+2} - \mathbf{u}_{t+1} \rangle + \gamma_{t+2} \langle \mathbf{b}_{t+1}, \widetilde{\mathbf{v}}_{t+2} - \mathbf{v}_{t+1} \rangle$$

$$+ \frac{\beta_\lambda}{2} \gamma_{t+2}^2 (\|\widetilde{\mathbf{u}}_{t+2} - \mathbf{u}_{t+1}\|^2 + \|\widetilde{\mathbf{v}}_{t+2} - \mathbf{v}_{t+1}\|^2)$$

$$\leq \mathcal{L}(\mathbf{u}_{t+1}, \mathbf{v}_{t+1}, \mathbf{w}_{t+1}) + \gamma_{t+2} \|\mathbf{a}_{t+1}\| (\rho + \rho') + \frac{\beta_\lambda}{2} \gamma_{t+2}^2 (\zeta^2) \tag{16}$$

Rearrranging terms we get

$$- \mathcal{L}(\mathbf{u}_{t+1}, \mathbf{v}_{t+1}, \mathbf{w}_{t+1}) \leq - \mathcal{L}(\mathbf{u}_{t+2}, \mathbf{v}_{t+2}, \mathbf{w}_{t+1}) + \gamma_{t+2} \|\mathbf{a}_{t+1}\| (\rho + \rho') + \frac{\beta_\lambda}{2} \gamma_{t+2}^2 \zeta^2 \tag{17}$$

Substituting Eqn. (17) in Eqn. (15), we get

$$\Delta_{t+1} - \Delta_t \leq - \left( \frac{2\eta_t}{\lambda} \right) \left( \mathcal{L}(\mathbf{u}_{t+2}, \mathbf{v}_{t+2}, \mathbf{w}_{t+1}) - \mathcal{L}(\widehat{\mathbf{u}}_{t+1}, \widehat{\mathbf{v}}_{t+1}, \mathbf{w}_{t+1}) - \frac{4\eta_t}{\lambda} \|\mathbf{a}_{t+1}\| (\rho + \rho') - \frac{8\eta_t^2 \beta_\lambda \zeta^2}{\lambda^2} \right)$$

$$+ \frac{4\eta_t}{\lambda} \|\mathbf{a}_{t+1}\| (\rho + \rho') + \beta_\lambda \frac{8\eta_t^2}{\lambda^2} \zeta^2 - \frac{\eta_t \alpha^2}{2L_\lambda \zeta^2} \min \left\{ \Delta_{t+1}^{(\mathrm{d})}, \frac{L_\lambda \zeta^2}{2} \right\}$$

$$= - \left( \frac{2\eta_t}{\lambda} \right) \Delta_{t+1}^{\mathrm{p}} - \frac{\eta_t \alpha^2}{2L_\lambda \zeta^2} \min \left\{ \Delta_{t+1}^{(\mathrm{d})}, \frac{L_\lambda \zeta^2}{2} \right\} + \left( \frac{4\eta_t}{\lambda} \|\mathbf{a}_{t+1}\| (\rho + \rho') + \beta_\lambda \frac{8\eta_t^2}{\lambda^2} \zeta^2 \right) \left( 1 + \frac{2\eta_t}{\lambda} \right)$$

$$\leq - \chi \eta_t \Delta_{t+1}^{\mathrm{p}} - \chi \eta_t \min \left\{ \Delta_{t+1}^{(\mathrm{d})}, \frac{L_\lambda \zeta^2}{2} \right\} + \left( \frac{4\eta_t}{\lambda} \|\mathbf{a}_{t+1}\| (\rho + \rho') + \beta_\lambda \frac{8\eta_t^2}{\lambda^2} \zeta^2 \right) \left( 1 + \frac{2\eta_t}{\lambda} \right)$$

$$\leq - \chi \eta_t \min \left\{ \Delta_{t+1}, \frac{L_\lambda \zeta^2}{2} \right\} + \left( \frac{4\eta_t}{\lambda} \|\mathbf{a}_{t+1}\| (\rho + \rho') + \beta_\lambda \frac{8\eta_t^2}{\lambda^2} \zeta^2 \right) \left( 1 + \frac{2\eta_t}{\lambda} \right)$$

where $\chi = \min \left\{ \frac{2}{\lambda}, \frac{\alpha^2}{2\beta_\lambda \zeta^2} \right\}$. Letting $\eta_t = \frac{2}{\chi(t+2)}$, we get

$$\Delta_{t+1} - \Delta_t \leq - \frac{2}{t+2} \min \left\{ \Delta_{t+1}, \frac{L_\lambda \zeta^2}{2} \right\} + \left( \frac{8}{\chi \lambda (t+2)} \|\mathbf{a}_{t+1}\| (\rho + \rho') + 32 \frac{\beta_\lambda \zeta^2}{\chi^2 \lambda^2 (t+2)^2} \right) \left( 1 + \frac{4}{\chi \lambda (t+2)} \right)$$

We thus have that,

$$\Delta_{t+1} - \Delta_t \leq - \frac{2}{t+2} \min(\Delta_{t+1}, \theta_1) + \frac{\theta_2}{(t+2)^2} + \frac{\theta_3}{t+2} (\rho + \rho') \tag{18}$$

where $\theta_1 = \frac{L_\lambda \zeta^2}{2}$, $\theta_2 = 32 \frac{\beta_\lambda \zeta^2}{\chi^2 \lambda^2} \left( 1 + \frac{2}{\chi \lambda} \right)$ and $\theta_3 = \frac{8}{\chi \lambda} \left( 1 + \frac{2}{\chi \lambda} \right) \max_t \|\mathbf{a}_{t+1}\|$. $\qquad \square$

From Lemma 8, we have that $\|\mathbf{w}_t\|$ is bounded by a constant if the duality gap $\Delta_t$ is bounded, and hence $\|\mathbf{a}_{t+1}\| = \|\nabla_{\mathbf{u}}\mathcal{L}(\mathbf{u}_{t+1}, \mathbf{v}_{t+1}, \mathbf{w}_{t+1})\| = \|\nabla\psi(\mathbf{u}_{t+1}) + \mathbf{w}_{t+1} + \lambda(\mathbf{u}_{t+1} - \mathbf{v}_{t+1})\|$ can also be bounded by a constant. We will need $2\theta_1 > \theta_3(\rho + \rho')$ for there to be a decrease in $\Delta_t$, this can be simply achieved by setting $\lambda$ to be a large enough value. Because if $\lambda$ is large, $\chi \approx \frac{c}{\lambda}$, and hence $\theta_3$ becomes a constant when increasing $\lambda$ further, but $\theta_1$ keeps increasing linearly with $\lambda$.

**Lemma 14.** *Let $\Delta_t$ be a sequence satisfying Eqn. (18). Let $2\theta_1 > \theta_3(\rho + \rho')$. Let there exist a $t_0 > \frac{\theta_2}{2\theta_1 - \theta_3(\rho + \rho')} - 2$ such that $\Delta_{t_0} \leq \theta_1$, then*

$$\Delta_t \leq \min\left\{\frac{4\theta_1(t_0 + 2)}{t + 2} + \frac{\theta_3(\rho + \rho')}{2}, \ \theta_1\right\} \quad \forall t \geq t_0 . \tag{19}$$

*Proof.* For $t = t_0$ the bound on $\Delta_t$ simplifies to $\theta_1$ and hence is true. This will form our base case for proof by induction. We make the induction assumption that for a $t \geq t_0$, $\Delta_t \leq \min\left\{\frac{4\theta_1(t_0 + 2)}{t + 2} + \frac{\theta_3(\rho + \rho')}{2}, \ \theta_1\right\}$.

If $\Delta_{t+1} > \theta_1$, then

$$\theta_1 < \Delta_{t+1} \leq \Delta_t - \frac{2\theta_1}{t + 2} + \frac{\theta_2}{(t + 2)^2} + \frac{\theta_3}{t + 2}(\rho + \rho')$$

$$\Delta_{t+1} \leq \theta_1 - \frac{2\theta_1}{t + 2} + \frac{\theta_2}{(t + 2)^2} + \frac{\theta_3}{t + 2}(\rho + \rho')$$

$$2\theta_1 - \theta_3(\rho + \rho') < \frac{\theta_2}{t + 2}$$

$$t < \frac{\theta_2}{2\theta_1 - \theta_3(\rho + \rho')} - 2$$

which contradicts $t \geq t_0 > \frac{\theta_2}{2\theta_1 - \theta_3(\rho + \rho')} - 2$. Hence $\Delta_{t+1} < \theta_1$. Thus, from Eqn. (18), we have

$$\Delta_{t+1} \leq \Delta_t - \frac{2}{2 + t}\Delta_{t+1} + \frac{\theta_2}{(t + 2)^2} + \frac{\theta_3}{t + 2}(\rho + \rho')$$

$$\Delta_{t+1}\frac{t + 4}{t + 2} \leq \Delta_t + \frac{\theta_2}{(t + 2)^2} + \frac{\theta_3}{t + 2}(\rho + \rho')$$

$$\Delta_{t+1}\frac{t + 4}{t + 2} \leq \frac{4\theta_1(t_0 + 2)}{t + 2} + \frac{\theta_3(\rho + \rho')}{2} + \frac{\theta_2}{(t + 2)^2} + \frac{\theta_3}{t + 2}(\rho + \rho')$$

$$\Delta_{t+1} \leq \frac{4\theta_1(t_0 + 2)}{t + 4} + \frac{\theta_3(\rho + \rho')}{2}\left(\frac{t + 2}{t + 4} + \frac{2}{t + 4}\right) + \frac{\theta_2}{(t + 2)(t + 4)}$$

$$\Delta_{t+1} \leq \frac{4\theta_1(t_0 + 2)}{t + 4} + \frac{\theta_3(\rho + \rho')}{2} + \frac{2\theta_1(t_0 + 2)}{(t + 2)(t + 4)}$$

$$\Delta_{t+1} \leq 4\theta_1(t_0 + 2)\left(\frac{1}{t + 4} + \frac{1}{2(t + 2)(t + 4)}\right) + \frac{\theta_3(\rho + \rho')}{2}$$

$$\Delta_{t+1} \leq 4\theta_1(t_0 + 2)\left(\frac{1}{t + 4} + \frac{1}{(t + 3)(t + 4)}\right) + \frac{\theta_3(\rho + \rho')}{2}$$

$$\Delta_{t+1} \leq 4\theta_1(t_0 + 2)\left(\frac{1}{t + 3}\right) + \frac{\theta_3(\rho + \rho')}{2}$$

And hence, we have

$$\Delta_{t+1} \leq \min\left\{\frac{4\theta_1(t_0 + 2)}{t + 3} + \frac{\theta_3(\rho + \rho')}{2}, \theta_1\right\}$$

The Lemma thus holds by induction. $\qquad\square$

Now we have to show that in a constant number of iterations $t_0$ we can reach a point such that $\Delta_{t_0} \leq \theta$.

**Lemma 15.** *Let $\Delta_t$ be a sequence satisfying Eqn. (18). Let $2\theta_1 > \theta_3(\rho + \rho')$. Then there exists a constant $t_0 > \frac{\theta_2}{2\theta_1 - \theta_3(\rho + \rho')} - 2$ such that $\Delta_{t_0} \leq \theta_1$.*

*Proof.* Clearly, there must exist a $t_0$ such that $\Delta_{t_0} \leq \theta_1$, because $\frac{1}{t}$ is a divergent series, and $\frac{1}{t^2}$ is a convergent series and $\Delta_t$ is bounded below by 0.

The same argument can be used for saying that $\Delta_t$ drops below $\theta_1$ infinitely often, and hence there exists $t_0 > \frac{\theta_2}{2\theta_1 - \theta_3(\rho + \rho')} - 2$ such that $\Delta_{t_0} \leq \theta_1$.

Let $t_0$ be the first instant $t > \frac{\theta_2}{2\theta_1 - \theta_3(\rho + \rho')} - 2$ such that $\Delta_t \leq \theta_1$, clearly this $t_0$ can be upper bounded by a constant that depends only on the two numbers $2\theta_1 - \theta_3(\rho + \rho')$ and $\theta_2$. $\qquad\square$

Putting together Lemmas 13, 14 and 15, we get the following corollary.

**Corollary 16.** *Let $2\theta_1 > \theta_3(\rho + \rho')$. There exists a constant $t_0 > 0$ such that*

$$\Delta_t \leq \frac{4\theta_1(t_0 + 2)}{t + 2} + \frac{\theta_3(\rho + \rho')}{2} \quad \forall t \geq t_0 .$$

*where $\theta_1 = \frac{L_\lambda \zeta^2}{2}$ and $\theta_3 = \frac{8}{\chi\lambda}\left(1 + \frac{2}{\chi\lambda}\right) \max_t \|\mathbf{a}_{t+1}\|$.*

**Lemma 17.** *Let $2\theta_1 > \theta_3(\rho + \rho')$. Let $t_0 \in \mathbb{N}$ be as in Lemma 15. Let $\mathbf{u}_t, \mathbf{v}_t, \mathbf{w}_t$ be as in Algorithm 2. Then for all $T > 2t_0$ and $T > 10$, there exists a $t \in [T/2, T]$ such that*

$$\|\mathbf{u}_t - \mathbf{v}_t\|^2 \leq \frac{c}{T} + c(\rho + \rho')$$

*for some constant $c > 0$.*

*Proof.* Rewriting Eqn. (14) here, we have

$$\begin{aligned}
&\Delta_{t+1} - \Delta_t \\
&\leq \mathcal{L}(\mathbf{u}_{t+2}, \mathbf{v}_{t+2}, \mathbf{w}_{t+1}) - \mathcal{L}(\mathbf{u}_{t+1}, \mathbf{v}_{t+1}, \mathbf{w}_{t+1}) + \frac{2\eta_t}{\lambda}\left(\mathcal{L}(\mathbf{u}_{t+1}, \mathbf{v}_{t+1}, \mathbf{w}_{t+1}) - \mathcal{L}(\widehat{\mathbf{u}}_{t+1}, \widehat{\mathbf{v}}_{t+1}, \mathbf{w}_{t+1})\right) \\
&\quad - \eta_t \|\widehat{\mathbf{u}}_{t+1} - \widehat{\mathbf{v}}_{t+1}\|^2
\end{aligned}$$

With the above equation as the starting point and proceeding as we do in Lemma 13, we get the below inequality that is similar to Eqn. (15)

$$\begin{aligned}
\Delta_{t+1} - \Delta_t &\leq -\left(\frac{2\eta_t}{\lambda}\right)\left(\mathcal{L}(\mathbf{u}_{t+1}, \mathbf{v}_{t+1}, \mathbf{w}_{t+1}) - \mathcal{L}(\widehat{\mathbf{u}}_{t+1}, \widehat{\mathbf{v}}_{t+1}, \mathbf{w}_{t+1})\right) + \frac{4\eta_t}{\lambda}\|\mathbf{a}_{t+1}\|(\rho + \rho') + \beta_\lambda \frac{8\eta_t^2}{\lambda^2}\zeta^2 \\
&\quad - \eta_t \|\widehat{\mathbf{u}}_{t+1} - \widehat{\mathbf{v}}_{t+1}\|^2
\end{aligned}$$

Let $h_{t+1} = (\mathcal{L}(C[h_{t+1}], \mathbf{v}_{t+1}, \mathbf{w}_{t+1}) - \mathcal{L}(\widehat{\mathbf{u}}_{t+1}, \widehat{\mathbf{v}}_{t+1}, \mathbf{w}_{t+1})) \geq 0$. We then have

$$\frac{2}{\lambda}\left(h_{t+1} - 2\|\mathbf{a}_{t+1}\|(\rho + \rho')\right) + \|\widehat{\mathbf{u}}_{t+1} - \widehat{\mathbf{v}}_{t+1}\|^2 \leq \frac{\Delta_t - \Delta_{t+1}}{\eta_t} + \eta_t \frac{8\beta_\lambda \zeta^2}{\lambda^2} \qquad (20)$$

Let the dual step size $\eta_t = \frac{2}{\chi(t+2)}$. Let $\{w_t\}_{T/2}^T$ be a sequence of positive weights. We set $w_t = t - T/2$. Let $\tau_t = \frac{w_t}{\sum_{t=\frac{T}{2}}^T w_t} = \frac{2t - T}{(T/2)(T/2 + 1)}$ be the associated normalized weights. The convex combination of Eqn. (20)

with weights $\tau_t$ gives us

$$\sum_{t=T/2}^{T} \tau_t \left( \frac{2}{\lambda} \left( h_{t+1} - 2\|\mathbf{a}_{t+1}\|(\rho + \rho') \right) + \|\widehat{\mathbf{u}}_{t+1} - \widehat{\mathbf{v}}_{t+1}\|^2 \right)$$

$$\leq \sum_{t=T/2}^{T} \tau_t \frac{\Delta_t - \Delta_{t+1}}{\eta_t} + \sum_{t=T/2}^{T} \tau_t \eta_t \frac{8\beta_\lambda \zeta^2}{\lambda^2}$$

$$= \frac{\tau_{T/2}}{\eta_{T/2}} \Delta_{T/2} - \frac{\tau_T}{\eta_T} \Delta_T + \sum_{t=T/2+1}^{T} \Delta_t \left( \frac{\tau_{t+1}}{\eta_{t+1}} - \frac{\tau_t}{\eta_t} \right) + \sum_{t=T/2}^{T} \tau_t \eta_t \frac{8\beta_\lambda \zeta^2}{\lambda^2}$$

$$\leq \sum_{t=T/2+1}^{T} \left( \frac{4\theta_1(t_0 + 2)}{t+2} + \frac{\theta_3(\rho + \rho')}{2} \right) \left( \frac{\tau_{t+1}}{\eta_{t+1}} - \frac{\tau_t}{\eta_t} \right) + \sum_{t=T/2}^{T} \tau_t \eta_t \frac{8\beta_\lambda \zeta^2}{\lambda^2}$$

$$= \sum_{t=T/2+1}^{T} \left( \frac{4\theta_1(t_0 + 2)}{t+2} \right) \left( \frac{8t - 2T + 12}{\chi T(T+2)} \right) + \left( \frac{\theta_3(\rho + \rho')}{2} \right) \left( \frac{\tau_T}{\eta_T} \right) + \sum_{t=T/2}^{T} \tau_t \eta_t \frac{8\beta_\lambda \zeta^2}{\lambda^2}$$

$$\leq \sum_{t=T/2+1}^{T} \left( \frac{4\theta_1(t_0 + 2)}{T/2+2} \right) \left( \frac{8T - 2T + 12}{\chi T(T+2)} \right) + \left( \frac{\theta_3(\rho + \rho')}{2} \right) \left( \frac{4(T+2)}{T2\chi} \right) + \sum_{t=T/2}^{T} \tau_T \eta_{T/2} \frac{8\beta_\lambda \zeta^2}{\lambda^2}$$

$$\leq \sum_{t=T/2+1}^{T} \left( \frac{8\theta_1(t_0 + 2)}{T} \right) \left( \frac{12}{\chi T} \right) + \left( \frac{\theta_3(\rho + \rho')}{2} \right) \left( \frac{8}{2\chi} \right) + \sum_{t=T/2}^{T} \frac{4}{T} \frac{4}{\chi T} \frac{8\beta_\lambda \zeta^2}{\lambda^2}$$

$$\leq \frac{48\theta_1(t_0 + 2)}{\chi T} + \frac{2\theta_3(\rho + \rho')}{\chi} + \frac{64\beta_\lambda \zeta^2}{\chi \lambda^2 T}$$

Thus, there must exist a $t \in [T/2, T]$ such that

$$\frac{2h_{t+1}}{\lambda} + \|\widehat{\mathbf{u}}_{t+1} - \widehat{\mathbf{v}}_{t+1}\|^2 \leq \frac{48\lambda^2 \theta_1(t_0 + 2) + 64\beta_\lambda \zeta^2}{\chi \lambda^2 T} + \left( \frac{2\theta_3}{\chi} + \frac{4}{\lambda} \max_t \|\mathbf{a}_t\| \right) (\rho + \rho'). \quad (21)$$

Now from Lemma 10 we have

$$\|(\mathbf{u}_{t+2} - \mathbf{v}_{t+2}) - (\widehat{\mathbf{u}}_{t+1} - \widehat{\mathbf{v}}_{t+1})\|^2 \leq \frac{2}{\lambda} \left( \mathcal{L}(\mathbf{u}_{t+2}, \mathbf{v}_{t+2}, \mathbf{w}_{t+1}) - \mathcal{L}(\widehat{\mathbf{u}}_{t+1}, \widehat{\mathbf{v}}_{t+1}, \mathbf{w}_{t+1}) \right) = \frac{2}{\lambda} \Delta_t^{(\mathrm{p})} \quad (22)$$

From Eqn. (16), we have

$$\Delta_{t+1}^{(\mathrm{p})} - h_{t+1} = \mathcal{L}(\mathbf{u}_{t+2}, \mathbf{v}_{t+1}, \mathbf{w}_{t+1}) - \mathcal{L}(\mathbf{u}_{t+1}, \mathbf{v}_{t+1}, \mathbf{w}_{t+1})$$

$$\leq \frac{4\eta_t}{\lambda} \|\mathbf{a}_{t+1}\|(\rho + \rho') + \frac{32\beta_\lambda \zeta^2}{\lambda^2 \chi^2 (t+2)^2} \quad (23)$$

Putting Eqns. (21), (22) and (23), we get Thus, there must exist a $t \in [T/2, T]$ such that

$$\|\mathbf{u}_t - \mathbf{v}_t\|^2 \leq \frac{48\lambda^2 \theta_1(t_0 + 2) + 64\beta_\lambda \zeta^2 + 16\lambda \max_t \|\mathbf{a}_t\|(\rho + \rho')}{\chi \lambda^2 T} + \left( \frac{2\theta_3}{\chi} + \frac{4}{\lambda} \max_t \|\mathbf{a}_t\| \right)(\rho + \rho') + O\left( \frac{1}{T^2} \right).$$
$$(24)$$
$\square$

# B  Further Related Work

## B.1  The COCO Approach

We elaborate on the COCO approach of Narasimhan (2018) [29], which is statistically consistent, but is shown to achieve only a $O(1/\epsilon^3)$ convergence rate. Like us, this approach also reformulates (OP1) as an optimization over $\mathcal{C}$ but retains explicit constraints $\phi_k(C) \leq 0, \forall k$:

$$\min_{C \in \mathcal{C}} \psi(C)$$
$$\text{s.t. } \phi_k(C) \leq 0, \forall k \in [K].$$

The idea is to then formulate the Lagrangian for the constrained problem with Lagrange multipliers $\lambda \in \Lambda \subset \mathbb{R}_+^K$:

$$\mathcal{L}(C, \lambda) = \psi(C) + \sum_{k=1}^{K} \lambda_k \phi_k(C),$$

and to maximize the Lagrangian over the multipliers using gradient ascent:

$$C^{(t+1)} \in \underset{C \in \mathcal{C}}{\operatorname{argmin}} \mathcal{L}(C, \lambda^{(t)}) \tag{25}$$

$$\lambda^{t+1} = \Pi_\Lambda \left( \lambda^{(t)} - \nabla_\lambda \mathcal{L}(C^{(t+1)} \lambda^{(t)}) \right), \tag{26}$$

where $\Pi_\Lambda$ is the projection onto the set $\Lambda$.

Note, however, that each gradient update on $\lambda$ requires a minimization of the Lagrangian over $\mathcal{C}$ in (25), and this is performed with a full run of the classical Frank-Wolfe method [18] using calls to a plug-in routine to solve the LMO needed in each iteration. The final algorithm has two levels of nesting, where the inner level solves the minimization in (25) with $O(1/\epsilon)$ calls to the plug-in routine, and the outer level performs $O(1/\epsilon^2)$ gradient ascent steps, resulting in a total $O(1/\epsilon^3)$ calls to the plug-in routine to reach an $\epsilon$-optimal, $\epsilon$-feasible solution.

## B.2 The 3-player Approach

As noted in Section 1, Narasimhan et al. (2019) [30] provide an idealized algorithm that enjoys the same convergence rate as our approach to the optimal feasible solution, but do not provide a full-fledged consistency analysis for this method. We elaborate on this method below.

Under the assumption that $\psi$ and $\phi_k$'s are monotonically non-decreasing in their arguments, Narasimhan et al. reformulate (OP1) by introducing slack variables $\xi \in [0,1]^d$ and arrive at the following equivalent problem:

$$\min_{h \in \mathcal{H}, \, \xi \in [0,1]^d} \psi(\xi)$$
$$\text{s.t. } \phi_k(\xi) \le 0, \, \forall k \in [K]$$
$$\xi \ge C[h]$$

They then formulate the Lagrangian for this problem with multipliers $\lambda \in \Lambda \subset \mathbb{R}_+^{K+d}$:

$$\mathcal{L}(h, \xi, \lambda) = \psi(\xi) + \sum_{k=1}^{K} \lambda_k \phi_k(\xi) + \sum_{i=1}^{d} \lambda_{K+i} \left( C_i[h] - \xi_i \right),$$

and perform the following sequence of updates at each step $t$:

$$h^{(t+1)} \in \underset{h \in \mathcal{H}}{\operatorname{argmin}} \mathcal{L}(h, \xi^{(t)}, \lambda^{(t)}) \tag{27}$$

$$\xi^{(t+1)} \in \underset{\xi \in [0,1]^d}{\operatorname{argmin}} \mathcal{L}(h^{(t)}, \xi, \lambda^{(t)}) \tag{28}$$

$$\lambda^{t+1} = \Pi_\Lambda \left( \lambda^{(t)} - \nabla_\lambda \mathcal{L}(h^{(t+1)}, \xi^{(t+1)} \lambda^{(t)}) \right), \tag{29}$$

where $\Pi_\Lambda$ is the projection onto the set $\Lambda$.

The authors then show that when $\psi$ and $\phi_k$'s are convex, these updates converge to an $\epsilon$-optimal, $\epsilon$-feasible classifier after $O(1/\epsilon^2)$ steps. However, this result relies on access to an oracle for performing the optimization in (27) over the space of classifiers $\mathcal{H}$ near-optimally. The authors further acknowledge that such an oracle may not exist for general settings, and prescribe a more 'practical' algorithm that replaces (27) with a gradient update on a relaxed Lagrangian objective, but does not enjoy the same convergence guarantees. We compare against this surrogate-based approach (referred to as the 3-player method) in the experiments in Section 5. In the open-source implementation the authors provide [10], they further replace (28) with a gradient update on $\xi$.

The algorithm we propose in this paper also uses two minimization subroutines, namely an LMO over $\mathcal{C}$ and an LMO over $\mathcal{F}$, but both of these can be implemented efficiently. The LMO over $\mathcal{C}$ is implemented in our approach using a plug-in classifier, and the LMO over $\mathcal{F}$ reduces to a simple convex program and can often be implemented very efficiently with a specialized solver. Unlike the the 3-player approach, we do not maintain an explicit Lagrange multiplier for each constraint, and access the constraint set only through an LMO. In Section 4, we then provide a complete consistency analysis, showing optimality and feasibility bounds in terms of the quality of class-probability estimates used to implement the plug-in classifier. Our results, however, require the objective function $\psi$ to be smooth, whereas Narasimhan et al. do not require this.

Table 4: Datasets used in our experiments.

| Dataset | Instances | Features | Protected Attribute | Prot. group frac. |
|---|---|---|---|---|
| COMPAS | 6172 | 32 | Gender | 0.19 |
| Communities & Crime | 1994 | 132 | Race | 0.49 |
| Law School | 20798 | 16 | Race | 0.06 |
| Adult | 48842 | 123 | Gender | 0.10 |
| Default | 30000 | 23 | Gender | 0.40 |

(a) Default

(b) Law School

(c) Adult

(d) Crime

Figure 4: Training objective (left) and constraint violation (right) as function of no. of plug-in calls for the task of minimizing G-mean subject to an equal opportunity constraint.

## C  Additional Experimental Details

Table 4 lists the datasets used in our experiments. Figure 4 shows convergence of the proposed method and the prior COCO method as function of the number of calls to the plug-in method. Figure 5 demonstrates robustness of our approach to hyper-parameter choices.

**Hyper-parameters.** To implement the plug-in routine, we use a pre-trained linear logistic regression model to estimate $\widehat{\boldsymbol{\eta}}$, with the protected attribute included as one of the features. For the problems we consider in the experiments, the LMO over the feasible set $\mathcal{F}$ in the proposed SBFW method is a linear program (LP), which we solve using a standard LP solver. For the proposed SBFW, we set $\eta_t$ as a decreasing step function with values $\{0.5, 0.1, 0.001\}$, set $\lambda = 10$ and $\gamma_t = \frac{2}{t+2}$. For COCO, we tuned the learning rates from the range $\{0.01, 0.1, 0.5, 1, 10, 20\}$. For the 3-player approach, we tuned the learning rates for the model and constraint from $\{0.01, 0.1, 0.5, 1\}$. We note that SBFW requires almost no tuning compared to COCO and 3-player for the experiments. In each case, we pick the best hyper-parameter using a heuristic provided by Cotter et al. (2019) [10] to find the best trade-off between the training objective and constraint violations. We ran the 3-player

Figure 5: Robustness to hyper-parameters: Scatter plot of train G-mean and equal opportunity violation (with negative values clipped to zero) for six step sizes. While all six choices achieved close-to-best objective and near-zero violations for the proposed algorithm, only two choices led to similar metrics for COCO, and three choices led to similar metrics for the 3-player method.

Figure 6: An illustration of Algorithm 1 for a synthetic 2-class problem with 20 constraints. We consider a data distribution with equal prior probabilities and with class conditionals $X|Y=0$ and $X|Y=1$ distributed as a standard normal with means 1 and 0 respectively. The goal is to minimize H-mean subject to a 20-sided polygonal constraint.

method for 2000 iterations and ran COCO and SBFW with 2000 calls to the plug-in routine. In the experiments in Figure 4, we separately tune the hyper-parameters for each method. For all experiments, we measure the constraint violation by the positive part of $\phi(h) - \epsilon$, that is using $\max\{0, \phi(h) - \epsilon\}$.

**Larger number of constraints.** In our final experiment, we demonstrate the effectiveness of the proposed approach in handling a larger number constraints than considered in Section 5. For this, we consider a synthetic 2-class problem, with equal prior probabilities and with class conditionals $X|Y=0$ and $X|Y=1$ distributed as a standard normal with means $+1$ and 0 respectively. The goal is to minimize H-mean subject to the diagonal entries of the confusion matrix that the confusion matrix lies within a polygon centered at $(0.2, 0.2)$. Figure 6 shows that contours of the objective function and the polygonal constraint region highlighted in red. The polygon is represented by 20 linear constraints. We find that the proposed method converges to a near-optimal, near-feasible solution within 40 calls to the plug-in routine.