[Reviews · NeurIPS 2020]

Review 1

Summary and Contributions: In this paper, the authors present an algorithm for constrained (multi-class) classification problems where both the objective and the constraints depend on the confusion matrix. In particular, the results of the paper may be applied to fair multi-class problems. The authors present an algorithm inspired from both augmented Lagrangian and Frank Wolfe algorithms. Since one of the two linear minimizations is not directly possible, the authors use plug-in estimators to circumvent this issue. For a consistent plug-in estimator, the authors show that the output given by the algorithm converges to the optimal solution (when the number of sample is infinite). The theoretical result is accompanied with empirical results. UPDATE: I would like to thank the authors for their feedback. I keep thinking that the paper deserves to be accepted for publication. However, note that the statistical results can be improved (see reviews).

Strengths: The paper widely improve the results from "Learning with complex loss functions and constraints": 1) It holds under weaker assumption 2) The dependence with respect to the number of iteration is improved 3) The dependence with respect to the number of constraints is also improved. Moreover, the result is very general and can be applied to many multi-class classification problems. The empirical results highlight all these improvements.

Weaknesses: The paper is a little bit abstract. It could be nice to see if all the assumptions of the paper are verified for usual problems (fairness for example). For example, is it the case that $\psi$ is Lipschitz and bounded ? It is very clear, but it would be even clearer with the precise presentation of an example.

Correctness: There is a mistake in the main Theorem: $\rho$ should be $\sqrt \rho$ (see Theorem supplementary material). It's minor and is just a typo I guess !

Clarity: The paper is VERY clearly written. Everything is fluent and easy to follow. It was a great pleasure to review this article !

Relation to Prior Work: I am not an expert of the field, but the bibliography looks complete and well-organize.

Reproducibility: Yes

Additional Feedback: TYPOS: -Line 90: a space is missing -Line 167: 1 should be $j$ in the conditional probability -Line 211: problem in this the second limit. It should not be 0. COMMENTS : Here are some (minor) details that could improve the clarity of the paper. 1) It could be nice to add the proof of proposition 2 2) Since the parameter $\lambda$ seems to be important (see the dependence in the main theorem), it would be nice to write the augmented Lagrangian with an index depending on $\lambda$. Could you precise what is the choice of lambda ? Normally, there should be a lower bound on $\lambda$. The main theorem should hold for any $\lambda \geq \bar \lambda$, with a specific $\bar \lambda$ 3) In the main Theorem: it would be nice to explicit the constant $\bar \epsilon$ 4) In the main theorem: say with respect to which random variables the expectation is taken in line 199 5) Put the definition of $Delta_n$ (the simplex in R^n) at least say it.


Review 2

Summary and Contributions: The authors look at multi-class classification problems under constraints depending (via a convex function) on general functions of the (ideal) confusion matrix, where the learning algorithm is assumed to output a stochastic classifer. Essentially, they break down the problem into two sub-tasks, related to the primary objective (also convexified here) and constraints, respectively, and working this sub-tasks into a linear objective. This formulation makes it reasonable to use traditional Frank-Wolfe type updates, and starting from a pre-trained base classifier (randomized), they use this update strategy to iteratively ensure the resulting (randomized) classifier does a “good” job (in a relative sense) of satisfying the constraints while still similarly achieving “good” performance in the primary objective. They give a procedure which is somewhat abstract, but one expects it can readily be implemented by users who flesh out the key abstracted components (sufficient stats, pre-trained classifier, linear minimization sub-routine). The authors give error bounds for this general-purpose procedure, depending on the approximation error of the pre-trained classifier and statistical error of typical empirical estimates in their sub-routines. The exposition comes with a fairly solid discussion of their algorithm design, and empirical tests are conducted to evaluate a practical implementation of their high-level procedure.

Strengths: The problem of interest is clear (if somewhat abstract), and the algorithm proposed here is implementable, grounded in clear computational and statistical principles, and comes with both theoretical and empirical analyses, yielding initial evidence for the efficacy of their approach, claiming that it is an order of magnitude more efficient (in terms of oracle calls) than a closely related prior work.

Weaknesses: The main limitation of the proposal here is that it is just a general-purpose wrapper around a pre-fixed classifier; in particular, this means that the actual quality of the output depends entirely on the quality of this base classifier, and that for certain special cases (of classifiers, sufficient stats, and constraints) it is perfectly plausible that a more efficient approach tuned to the setting could be found. This last point is the price that is paid for generality, of course.

Correctness: For the most part, the paper appears correct; however there is a clear issue with the statistical error term of the underlying procedure. The statistical error (the sqrt(log(1/delta)/N) term) is stated so roughly as to be on the verge of being patently false, as dimension dependence is completely hidden in the big-O notation. The authors’ procedure requires estimating the mean of a d-dimensional random vector, at each step. The authors casually mention (in the appendix) that a uniform version of Hoeffding’s inequality can be made with proper capacity notions, but that this is “not significant.” While somewhat technical, hiding the dependence on d (an important parameter from the perspective of the user constructing the performance metrics) is misleading.

Clarity: The main exposition and layout of the paper is fairly clear.

Relation to Prior Work: Prior work is discussed clearly.

Reproducibility: Yes

Additional Feedback: For the most part the paper is well-written, but I think that it should be a priority of the authors to present their main statistical result in the most clear and genuine way possible. The current form hides dependence on the dimension of the sufficient statistics and the iterative nature of the process (as mentioned above), and I believe needs revision. UPDATE: Having seen the feedback, I maintain my initial assessment, including the issues raised above.


Review 3

Summary and Contributions: The paper proposes a novel approach to learning a multiclass classifier under structural constraints motivated from fairness applications. The performance of a candidate classifier~$h$ -- a general mapping of the feature space on a probability simplex -- is measured through the so-called ``confusion matrix'' $C[h]$ whose vectorization stacks the vector of expected sufficient statistics extracted from a datapoint $(X,Y)$ and the classifier output~$\hat Y$. The goal is then to solve a convex program, where a smooth and convex los is minimized over the set of achievable confusion matrices, corresponding to the fixed (and unknown to the learner) population distribution, under a number of (convex) functional constraints. This setup has been earlier considered by Narasimkhan (2018). The authors advocate the following approach: the problem is cast as convex program with smooth objective, and with constraint set given as the intersection of the two sets (of confusion matrices): - the ``feasible'' set $\F$ corresponding to the functional constraints; - the ``achievable'' set $\C$ corresponding to all possible confusion matrices for the data-generating distribution. As such, only the set $\C$ depends on the unknown data-generating distribution. Following the approach in the earlier work of Narasimhan (N18), the authors then use that a linear minimization oracle for $\C$ corresponds to the Bayes-optimal predictor of the confusion matrix; thus one can approximate this oracle via the plug-in approach. Using this idea, N18 advocated an approach based on the Frank-Wolfe method. The departure of the present work from N18 is in how they the functional constraints are treated: instead of incorporating them into the Lagrangian, as done in N18, they use the augmented-Lagrangian type approach which leads to a better optimization complexity estimate.

Strengths: Theoretical analysis seems overall sound, although there is laxity in some of the derivations as I describe below. Another merit of the paper is a thorough experimental analysis; the results demonstrate the superiority of the proposed method to the baseline of N18 on a few datasets.

Weaknesses: The main issue I see with the paper is what I perceive as incremental nature of its contributions with respect to earlier work, notably Naramsimhan (2015; 2018-N18) and Gidel et al. (2018-G18). The connection between LMO for the set $\C$ and Bayesian predictor (Prop. 2 of the present paper, given without proof and explicit reference) has been first observed in N18; notably, nor N18 nor the present paper gives explicit proof. The sample complexity analysis for the plug-in approximation of LMO has been given in N18. In fact, to my (limited) understanding, essentially all components in the proof of Theorem 1 can already be found in existing works (N18 and G18). Besides, the notation closely follows that of N18 which I personally find a bit disturbing. Apart from these problems, I can see the following minor technical issue: as far as I can tell, the second bound in Lemma 1 of the supplementary must include a dimension-dependent factor. Indeed, the authors give this bound without a proof, with a brief informal discussion. Yet, from this very discussion I can see that: (1) even without covering the bound for the l2-norm is $O(\sqrt{d})$ larger than the coordinatewise bound; (2) covering might result in another~$O(\sqrt{d})$ factor. Perhaps the authours meant that the constants $c_1, c_2, ...$ are allowed to be dimension-dependent; in this case this could have been mentioned explicitly.

Correctness: Please see my previous answer.

Clarity: The clarity is overall reasonable but could have been improved. In particular, Proposition 2, given without proof, looks a bit cryptic. Mentioning ``standard uniform convergence argument'' in the proof of Lemma 1 is problematic (as I discussed in my previous answer).

Relation to Prior Work: The relevant work is mentioned and the contributions are positioned overall correctly with respect to it.

Reproducibility: Yes

Additional Feedback: UPDATE: In their rebuttal, the authors did address the minor of two issues I mentioned, the one with the dimension-dependent ``constant'' factor. Hence I'm improving my score from 5 to 6. I am still not convinced that technical contributions here are significant enough to be the key decision factor in whether the paper should be accepted. But the approach might be interesting to the broader community,


Review 4

Summary and Contributions: This paper presents a consistent algorithm for constrained classification problems where the constraints and objective are general functions of the confusion matrix. The proposed method reduces the constrained learning problem into a sequence of sub-problems that learns a plug-in classifier. The main contributions are a reduction in the number of calls to the plug-in sub-problem by formulating the learning task as optimization over the intersection of two convex sets.

Strengths: Compared to a previous but similar approach, the current method achieves a better convergence rate, extends to any non-smooth convex constraints (as opposed to requiring smooth and differentiable), and is able to take advantage of specialized solvers for optimization.

Weaknesses: Limitations of this work currently lie in the requirements of convex constraints, and also for the consistency proof, smoothness and Lipschitz are required. The empirical results don't appear to be largely better, motivating the question, is this paper's contribution mostly on a faster convergence rate?

Correctness: The claims appear to be correct

Clarity: For the most part, the paper reads well though there are a few typos/grammatical errors. Line 78 what is \Delta_n? It is present throughout the paper (in algorithm 2, line 175). Line 192, Is B(u, r) a ball with radius r?

Relation to Prior Work: Relation to prior work is clearly discussed, and builds off previous work requiring a larger number of calls to the plug-in learning routine.

Reproducibility: Yes

Additional Feedback: UPDATE: I have read the authors rebuttal, but will keep my original score unchanged.

[Author Response · NeurIPS 2020]

We greatly appreciate the reviewers for taking a close look at the paper and the proofs, and giving a detailed feedback. We first address the common concerns raised.

**Dependence on $d$ in Lemma 1:** While our focus has been on the dependence of our algorithm on the number of plug-in calls, we understand why the reviewers would like the dependence on $d$ to be made explicit. Below, we expand the statistical error term in Lemma 1 to show the dependence on $d$, and will include this in the paper along with the complete proof. This is the *same* dependence that the previous method of Narasimhan (2018) incurs [26].

For a $n$-class problem, let $\widehat{g}^{\mathbf{a}}, \widetilde{\mathbf{u}}^{\mathbf{a}} = \texttt{plug-in}(\mathbf{a})$ as in Algorithm 1. Then with probability $\geq 1 - \delta$ over draw of $N$ examples from the data distribution, we have for all $\mathbf{a} \in \mathbb{R}^d$:

$$||C[\widehat{g}^{\mathbf{a}}] - \widetilde{\mathbf{u}}^{\mathbf{a}}||_2 = O\left(d\sqrt{\frac{d\log(d) + \log(Nn^2) + \log(1/\delta)}{N}}\right)$$

where the notation $O$ only hides absolute constants. The proof follows from a straightforward application of a result from Cesa-Bianchi & Haussler (1998) to bound the growth function.

**Proof of Proposition 2:** Proposition 2 is straightforward and simply follows from expanding $\langle \mathbf{a}, C[h] \rangle$ as $\mathbf{E}_X\left[\sum_{y=1}^n \eta_y(X) \sum_{i=1}^d a_i \sigma_i(X, y, h(X))\right]$. Hence the Bayes-optimal classifier $h$ predicts for any given $x$, a label $\widehat{y}$ that minimizes the inner term $\sum_{y=1}^n \eta_y(x) \sum_{i=1}^d a_i \sigma_i(x, y, \widehat{y})$, i.e. $h(x) = \text{argmin}_{\widehat{y} \in [n]} \sum_{y=1}^n \eta_y(x) L_{y, \widehat{y}}(x)$. We'll definitely include this in the appendix.

**Reviewer 2: Lipschitzness.** The fairness and coverage constraints in Section 2 are Lipschitz in the confusion matrix, and so are the H-mean, Q-mean and Min-max metrics. For the G-mean and KLD metrics, we can easily construct close-approximations that are Lipschitz. We'll include these details in the paper, along with an example. As for the parameter $\lambda$, in theory it is sufficient to set it to a large-enough value as specified in Lemma 7. In practice, we set $\lambda = 10$, but the results were robust to changes in $\lambda$. Thanks for the suggestions to improve the writing and pointing out the typos!

**Reviewer 3: Limitations of a pre-fixed classifier.** We agree that the performance of a plug-in classifier depends on the quality of the base class probability model. As an alternative, one can always train a new classifier from scratch in each step of Algorithm 1 to solve the linear minimization (LMO) over $\mathcal{C}$ (line 5). This amounts to solving a cost-sensitive learning problem at each step. While the modified algorithm will be computationally more expensive, it no longer depends on a pre-trained model. Moreover, the number of calls to the LMO routine will be similar to Theorem 1, with the LMO-approximation term now depending on the quality of the classifier learned at each step. We'll include a discussion on this in the paper.

**Reviewer 4: Novelty.** While we agree that the paper combines ideas from prior works, our main contribution is the re-formulation of a constrained classification problem as an optimization problem over the *intersection of two sets* $\mathcal{C} \cap \mathcal{F}$, and the novel application of results from Gidel et al. (2018) [13] to solve the resulting optimization. This allows us to provide a new learning algorithm which (i) has a simpler structure than the previous algorithm, (ii) enjoys better convergence rate, (iii) can better handle non-smooth constraints, and (iv) is more robust to choices of hyper-parameters.

Moreover, the proofs don't directly follow from the previous papers for the following reasons: (i) Gidel et al. provide an optimization algorithm, which does not directly apply to a statistical ML setup. For example, their proofs assume an exact LMO, whereas we had to explicitly take into account the error due to finite sample, including in their so-called fundamental descent lemma. (ii) Gidel et al. only provide a bound on the duality gap for the constrained optimization problem; we convert this into a bound on the sub-optimality and infeasiblity of the learned classifier.

Finally, we are able to handle a broader class of learning problems than Narasimhan (2018) [26], where the performance metrics can be defined by functions of more general "confusion vectors", which can depend on the instance $x$ in more intricate ways.

**Reviewer 6: Convexity.** We require the objective and constraints to be convex in the confusion matrix. We don't see this as a strong requirement as it is satisfied by all the example metrics in Section 2, including common fairness metrics such as equal opportunity and equalized odds. Yes, $B(\mathbf{u}, r)$ is a ball of radius $r$ centered at $\mathbf{u}$; $\Delta_n$ is the $n$-dimensional simplex. We'll make these notations clear.

**Reference**

1. N. Cesa-Bianchi and D. Haussler. A graph-theoretic generalization of the Sauer-Shelah lemma. *Discrete Applied Mathematics*, 86(1):27–35, 1998.


[Meta-Review · NeurIPS 2020]

After reading the rebuttal, the reviewers reached the consensus of accepting the paper. Please do fix all the issues pointed out in the reviews, especially on making the dimension dependence explicit.